# Wring Out The Bias: A Rotation-Based Alternative To Projection Debiasing

**Walter Gerych**[1]*, **Cassandra Parent**[2], **Quinn Perian**[2], **Rafiya Javed**[3],
**Justin Solomon**[2], **Marzyeh Ghassemi**[2]
[1] WPI    [2] MIT    [3] Google

## Abstract

Vision-Language models (VLMs), including CLIP, are known to encode biases such as learning spurious correlations that falsely associate background attributes with particular labels. Debiasing approaches typically aim to isolate and remove subspaces corresponding to a target concept via projecting the embedding away from the concept. This strategy succeeds in debiasing VLM embeddings with respect to the concepts considered but can *amplify* biased shortcuts in unconsidered concepts. In practice, it is impossible to enumerate all possible biases, meaning that an increase in bias can go unobserved during evaluation. We propose a debiasing approach for a set of known concepts such that the relation to the remaining, *unconsidered*, concepts is minimally changed. We achieve this by rotating the VLM's embeddings within only a relevant subspace, rather than removing these subspaces, which mitigates unintended bias amplification.

## 1 Introduction

CLIP (Radford et al., 2021) and other Vision-Language models (VLMs) (Li et al., 2022; Liu et al., 2024) have revolutionized zero and few-shot learning. VLMs are now widely applied to a wide range of tasks, including image retrieval (Lahajal et al., 2024), image classification (Radford et al., 2021; An et al., 2023), and face recognition (Zhao and Patras, 2023). Despite this increase in popularity, VLMs consistently encode biases, leading to poor downstream performance (Alabdulmohsin et al., 2024; Kim et al., 2024; Friedrich et al., 2023). For example, VLMs are known to learn spurious correlations that associate background attributes with particular labels instead of learning from the object itself (Kim et al., 2024; Hamidieh et al., 2024).

VLM debiasing is an active area of research (Berg et al., 2022; Chuang et al., 2023a; Kong et al., 2024; Kim et al., 2024; Wang et al., 2021; Luo et al., 2024). Some approaches finetune the vision or language encoders of the VLM (Zhu et al., 2023; Alabdulmohsin et al., 2024; Shen et al., 2023), while others use post-processing to modify the embeddings directly (Jung et al., 2024; Gerych et al., 2024; Chuang et al., 2023a; Dehdashtian et al., 2023). Leading VLM debiasing approaches often rely on *projection* debiasing, which transforms the original embeddings to be orthogonal to a chosen concept direction (Wang et al., 2021; Chuang et al., 2023a; Seth et al., 2023; Zhang et al., 2024). Critically, we argue that while this approach effectively debiases against a chosen concept, it can *amplify* bias in unconsidered concepts. For instance, removing background information from the model's embeddings can result in the model relying even more heavily on biases associated with object type (Figure 1a).

This failure of projection debiasing approaches align with the known *whac-a-mole* dilemma (Li et al., 2023), where models that are debiased for one concept have amplifications in their remaining bias shortcuts. When this happens, the models are not really debiased — the bias is just transferred and hidden elsewhere. As more systems begin to rely on VLMs, it is impossible to consider all possible concepts during debiasing.

It is essential to develop debiasing approaches that can alter embeddings without amplifying bias in unconsidered concepts. However, avoiding bias amplification is challenging. There are innumerable *unconsidered concepts* for which labels are not available, but a successful method must ensure that

---

*Correspondence to `wgerych@wpi.edu`.

debiasing one concept does not significantly increase bias in any of these unconsidered concepts. Without labels for these concepts, we cannot *explicitly* optimize for this. Further, identifying the appropriate *directions or subspaces* in the VLM embeddings corresponding to a given concept is nontrivial, as concepts are often multifaceted and embedded in complex, high-dimensional spaces. Lastly, and crucially, the mechanism that causes bias amplification is not well understood.

In this work, we first identify a root cause of the bias-amplification "whac-a-mole" dilemma *for projection debiasing*. Specifically, we show that using projection to remove subspaces from VLM embeddings will *always* amplify biases in all orthogonal subspaces, and will cause unpredictable changes in bias for related subspaces. Next, we propose a simple yet effective alternative to projection: **W**eighted **R**otational Debias**ING** (WRING), a VLM debiasing approach designed to circumvent the whac-a-mole dilemma and avoid amplifying biases for unconsidered concepts. We show that this small change results in a debiasing approach that has *no* amplification of bias for orthogonal subspaces, and significantly mitigated changes in related subspaces.

A high level description of WRING is given as such: for a given embedding, WRING rotates the embedding such that the angle between each *group direction* and the embedding are equal to each other. These *group directions* are defined as axes in the VLM's embedding space that encode information about each *group*, where a *group* is a subclass of a given *concept* that we aim to debias for. For instance, for a dataset with images of dogs, one target concept is `dog breed` with {`labrador`, `pitbull`} groups. By ensuring equal cosine similarity between `labrador` and `pitbull`, we are ensuring that the embedding will not be more strongly associated with one group than the other.

**Contributions.** The primary contributions of this work are:

- We demonstrate that projection debiasing is prone to the whac-a-mole dilemma, meaning that biases in unconsidered concepts are amplified after projection.
- We propose WRING, the first VLM debiasing method designed to circumvent the whac-a-mole dilemma. WRING removes bias for known spurious concepts without amplifying biases for unconsidered concepts.
- We empirically demonstrate the effectiveness of WRING for debiasing known concepts without amplifying remaining biases in four datasets (2 datasets for racial and gender bias, 1 dataset of images of dogs with breed and background bias, 1 dataset of clothing images with color, season, and gender bias). In each dataset, we test our method using a different set of concepts.

## 2 RELATED WORK

**Bias in Vision-Language Models and Current Debiasing Approaches.** Despite Vision-Language Models becoming widespread (Radford et al., 2021; Ramesh et al., 2022; Saharia et al., 2022; Rombach et al., 2022), these models suffer from spurious correlations (Yang et al., 2023) and biases (Birhane et al., 2021; Zhang et al., 2022). These biases can originate at the dataset level (Birhane et al., 2021) and propagate to VLMs, amplifying societal biases (Zhang et al., 2022). For example, it was found that VLMs associate women with words like "shopping" and "cooking" whereas men are more associated with "working" and "driving" (Zhang et al., 2022). VLMs can also have background bias, where models learn spurious correlations from the background of an image that a user would consider unrelated to the given task (Kim et al., 2024; Hamidieh et al., 2024). There have been many attempts to debias VLMs including data augmentation and balancing (Bhargava and Forsyth, 2019), model-level adjustments such as adversarial training (Srinivasan and Bisk, 2021), and additive residual image representations (Seth et al., 2023).

**Projection-Based Debiasing.** A common assumption is that VLM models encode concepts linearly, meaning there are existing subspaces that correspond to distinct concepts (Chuang et al., 2023b; Bolukbasi et al., 2016; Alabdulmohsin et al., 2024; Papadimitriou et al., 2025). Thus, a standardized debiasing approach is to make an embedding orthogonal to the direction associated with the targeted spurious concept (Chuang et al., 2023b; Bolukbasi et al., 2016). Projection-based debiasing underlies many influential methods for mitigating bias in embedding models (Bolukbasi et al., 2016; Wang et al., 2021; Chuang et al., 2023a; Seth et al., 2023; Zhang et al., 2024). In this approach, bias is assumed to be localized within a low-dimensional subspace, and removing this subspace from the representation is intended to eliminate the associated unwanted information.

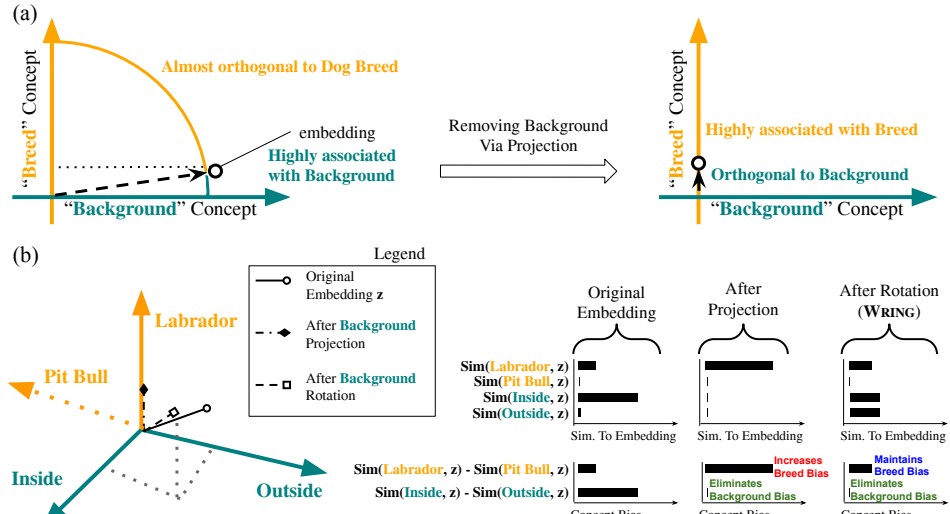

Figure 1: (a) An illustrative example of how using projection debiasing to remove one concept (`background`) can increase the bias/association with remaining concepts (`dog breed`). This example is a simplification of the actual case, as here we show one direction per `concept` whereas in reality there is one direction per `group` for all the groups on the `context`. This simplification is done for the purpose of 2D visualization. (b) An illustrative example of the difference between projection debiasing and our proposed rotational debiasing alternative. Consider `dog breed` = {`labrador`, `pit bull`} and `background` = {`inside`, `outside`}. The initial embedding is closer to `inside` and `labrador`, and orthogonal to the `pitbull` direction. When performing our proposed WRING debiasing, the embedding's similarity to `inside` becomes equal to its similarity to `outside`, likewise eliminating `bias`. Crucially, the similarity to `labrador` and `pit bull` is unchanged from the initial embedding, resulting in no amplification of `dog breed` bias.

We find bias amplification persists in current projection debiasing approaches. When projecting the embedding away from one concept, projection debiasing can alter the embeddings to be closer to another unconsidered concept. We describe the mechanisms causing this in the following section.

## 3 PROJECTION AMPLIFIES OTHER BIASES

VLMs such as CLIP (Radford et al., 2021) typically consist of an image encoder and a text encoder, and are trained to map image and text embeddings into an $n$-dimensional shared embedding space $\mathbb{V} \subseteq \mathbb{R}^n$. Given a dataset of paired images and texts — where a pair is often an image and its caption — VLMs are trained through contrastive learning to place paired image and text embeddings close together, and disparate instances further apart (Radford et al., 2021). This process aims to construct an embedding space where similar concepts have a high *cosine similarity*; e.g., images of pets will have a higher cosine similarity with each other than with images of plants or vehicles. However, due to spurious correlations in training data, this can lead to unfavorable results where concepts that a user would want to be unrelated are instead clustered together in the learned embedding space (Hamidieh et al., 2024).

Let $z \in \mathbb{V}$ be the embedding of a *query*, such as `"a photo of a dog"`, and we want to find image embeddings with a high similarity to the query embedding (e.g., for a retrieval task). Each image embedding has a corresponding *spurious attribute label* $c \in C$, such that the VLM has learned a spurious relationship between the concepts referenced in the query and the spurious attributes in $C$. For instance, the model may have learned a spurious correlation between `dog` and the background of the image, in which case the set of spurious attributes could be $C = \{$`indoors`, `outdoors`, ...$\}$.

Empirically, embedding models often encode concepts such that the variance of embeddings relating to the same concept falls mostly into a lower-dimensional subspaces (Bolukbasi et al., 2016). Prior work (Chuang et al., 2023a) has shown that the subspace for a spurious concept can be defined as

such: Let $\boldsymbol{A}_C \in \mathbb{R}^{n \times m}$ be a matrix with $m$ linearly independent columns $[\boldsymbol{c_1}, \boldsymbol{c_2}, \ldots, \boldsymbol{c_m}]$, where each $\boldsymbol{c_i} \in \mathbb{V}$ corresponds to the embedding of an element of the concept set $C$. Using the running example where the spurious concept is background with $C = \{\text{indoors}, \text{outdoors}, \ldots\}$, then $\boldsymbol{c_1}$ is the embedding for indoors, $\boldsymbol{c_2}$ is the embedding for outside, and so on. The subspace for $C$ is thus defined as $\text{col}(\boldsymbol{A}_C)$, the column space of $\boldsymbol{A}_C$. Alternatively, given a labeled dataset with (image, spurious concept label) pairs, each $\boldsymbol{c_i}$ can be defined as an average image embedding for images associated with the $i$th spurious attribute in $C$ (Gerych et al., 2024) — which we find to perform better in practice (see the Appendix).

We define bias between a query embedding $\boldsymbol{v}$ and spurious attribute embeddings $\boldsymbol{c_1}, \boldsymbol{c_2}$ as:

$$\text{bias}(\boldsymbol{v}, \boldsymbol{c_1}, \boldsymbol{c_2}) := \text{cosine\_sim}(\boldsymbol{v}, \boldsymbol{c_1}) - \text{cosine\_sim}(\boldsymbol{v}, \boldsymbol{c_2}).$$

This means the query $\boldsymbol{v}$ is biased towards $\boldsymbol{c_1}$ (e.g. indoors) over $\boldsymbol{c_2}$ (e.g. outdoors) when $\boldsymbol{v}$ has a higher relative similarity to INDOOR embeddings.

Projection-based debiasing approaches aim to correct this bias by making each query embedding $\boldsymbol{v}$ orthogonal to $\boldsymbol{A}_C$ (Bolukbasi et al., 2016):

$$\boldsymbol{v}_{\backslash C} = \boldsymbol{v} - \boldsymbol{P}_C \boldsymbol{v},$$

where $\boldsymbol{P}_C = A_C \left(A_C^\top A_C\right)^{-1} A_C^\top$ is the orthogonal projection matrix of $A_C$. This projection technique is simple and easy to integrate into other approaches, and has become a ubiquitous method of debiasing embedding models (Wang et al., 2021; Chuang et al., 2023a; Seth et al., 2023; Zhang et al., 2024). However, we argue that projection alters — and generally amplifies — biases for other, unconsidered concepts:

**Projection Changes Bias For Unconsidered Concepts.** Let $\boldsymbol{v}_{\backslash C}$ be the resulting embedding after making $\boldsymbol{v}$ orthogonal to $\text{col}(\boldsymbol{A}_C)$, the subspace corresponding to concept $C$. Let $\boldsymbol{d_1}, \boldsymbol{d_2} \in \text{col}(\boldsymbol{A}_D) \neq \text{col}(\boldsymbol{A}_C)$ be two embeddings in the subspace for concept $D \neq C$. The change in bias between $\boldsymbol{v}$ and $\boldsymbol{d_1}, \boldsymbol{d_2}$ is given by:

$$\underbrace{\text{bias}(\boldsymbol{v}_{\text{PROJECTION},C}, \boldsymbol{d_1}, \boldsymbol{d_2})}_{\text{bias after Projection}} = \underbrace{\frac{\|\boldsymbol{v}\|}{\|\boldsymbol{v} - \boldsymbol{P}_C \boldsymbol{v}\|}}_{\text{bias amplification}} \cdot \text{bias}(\boldsymbol{v}, \boldsymbol{d_1}, \boldsymbol{d_2}) + \underbrace{\frac{\Delta_{\boldsymbol{P}_C \boldsymbol{v}}}{\|\boldsymbol{v} - \boldsymbol{P}_C \boldsymbol{v}\|}}_{\text{bias altering}}, \qquad (1)$$

$$\Delta_{\boldsymbol{P}_C \boldsymbol{v}} = \frac{\|d_1\| \langle \boldsymbol{P}_C \boldsymbol{v}, \boldsymbol{d_2} \rangle - \|d_2\| \langle \boldsymbol{P}_C \boldsymbol{v}, \boldsymbol{d_1} \rangle}{\|\boldsymbol{d_1}\| \|\boldsymbol{d_2}\|}.$$

**Bias amplification.** The first term in Equation 1 shows that projection scales the initial bias by a constant bias amplification term, such that this term is $> 1$ (as $\|\boldsymbol{v} - \boldsymbol{P}_C \boldsymbol{v}\| < \|\boldsymbol{v}\|$). This acts to increase any existing biases for the unconsidered concept $D$.

**Bias altering term.** The second term in Equation 1 is the *bias altering term*, which further increases bias when $\Delta_{\boldsymbol{P}_C \boldsymbol{v}}$ is positive — which happens when $\boldsymbol{P}_C \boldsymbol{v}$, the component of $\boldsymbol{v}$ in the subspace that was projected out, is more similar to $\boldsymbol{d_2}$ than $\boldsymbol{d_1}$. When this relationship is reversed and $\Delta_{\boldsymbol{P}_C \boldsymbol{v}}$ is negative, then this term acts to reduce the bias — but will only lead to a net reduction in bias when the bias altering term is of sufficient magnitude to cancel out the effects of the bias amplification term.

**Consistent amplification when subspaces are orthogonal.** When $\text{col}\,\boldsymbol{A}_D \perp \text{col}\,\boldsymbol{A}_C$ (the subspaces for $C$ and $D$ are orthogonal), the bias altering term is 0 and thus projection will consistently amplify bias. In practice, we expect this to be often the case. Draws from random vectors in high-dimensional spaces will be orthogonal with high probability (Ball et al., 1997), and our empirical observations support this (see Appendix).

The fact that projection amplifies biases for remaining concept subspaces can be shown through simple linear algebra, and the preceding analysis was straight forward. But the implication is important: Projection generally *amplifies* bias for embeddings outside of the space that was specifically debiased for, and only decreases bias for these embeddings under certain circumstances. If we had knowledge

of every possible spurious concept, then we could explicitly debias for all of them. The issue is that in general it is impossible to enumerate through all concepts for which a bias might exist. **This means that using projection debiasing is a gamble: projection will alter — and often amplify — biases for unconsidered concepts**.

# 4 WRING ROTATION: AVOIDING UNCONSIDERED BIAS AMPLIFICATION

In the previous section we showed that projection debiasing changes (and often amplifies) biases for remaining, unconsidered concepts. We require an alternative to this approach, that mitigates this tendency to amplify biases.

In essence, WRING subtracts the biased subspace from the embedding $v$ and replaces it with a norm-preserving term $w$:

$$v_{\text{WRING},C} := v - P_C v + \|P_C v\| \cdot w, \tag{2}$$

where $w$ is a unit vector with the following two properties: 1) $w$ is a vector in the concept subspace for $C$, $w \in \text{col}(A_C)$. 2) $w$ has an equal cosine similarity to each spurious attribute embedding, where each attribute embeddings $c_i$ is a column of $A_C$; $\text{bias}(w, c_i, c_j) = 0 \ \forall \ i, j$.

Property 1) ensures that $\|v_{\text{WRING},C}\| = \|v\|$, and that the angles between any embeddings orthogonal to $\text{col}(A_C)$ are unchanged — and thus bias for a concept unrelated to $C$ is unchanged. Property 2) makes $v_{\text{WRING},C}$ have an equal relation to every spurious attribute in $C$, and thus not be more biased towards any of them. For example, if $C = \{\texttt{indoors}, \texttt{outdoors}, \ldots\}$, then $v_{\text{WRING},C}$ would have an equal association with the embedding of $\texttt{indoors}$ and $\texttt{outdoors}$. The unique up to scale representation that satisfies our formulation is given by:

$$\tilde{w} = A_C \left( A_C^\top A_C \right)^{-1} \mathbf{1}.$$

We derive $\tilde{w}$ in the Appendix.

**WRING Improves The Limitations of Projection:** Let $v_{\text{WRING},C}$ be the result of debiasing $v$ with WRING. Let $d_1, d_2 \in \text{col}(A_D) \neq \text{col}(A_C)$ be two embeddings in the subspace for concept $D \neq C$. The change in bias between $v$ and $d_1, d_2$ is given by:

$$\underbrace{\text{bias}(v_{\text{WRING},C}, d_1, d_2)}_{\text{bias after WRING}} = \text{bias}(v, d_1, d_2) + \frac{\|v - P_C v\|}{\|v\|} \underbrace{\frac{\Delta_{P_C v}}{\|v - P_C v\|}}_{\text{bias altering}} - \underbrace{\Delta_w}_{\text{dampening}} \tag{3}$$

where $\Delta_w = \left( \langle \hat{w}, d_j \rangle - \langle \hat{w}, d_i \rangle \right) / \left( \|v\| \|d_1\| \|d_2\| \right)$, $\hat{w} = |P_C v| w$.

**No bias amplification term.** Unlike the change in bias after projection (Equation 1), the $\text{bias}(v, d_1, d_1)$ term is not multiplied by any bias amplification term.

**Mitigated bias altering term.** Equation 3 shares the *bias altering term* with Equation 1. However, in Equation 3 this term is mitigated by scaling it by a multiplicative factor that is less than 1. A nice property of WRING is that when $\Delta_{P_C v} > 0$, this term *immediately* helps to reduce bias without the need to overcome the amplification factor on the first term.

**Dampening term.** Equation 3 includes the bias dampening term $\Delta_w$, which has a sign opposite to the bias amplification term. It acts as a mitigating factor and reduces the bias altering term.

**No amplification when subspaces are orthogonal.** When $\text{col } A_D \perp \text{col } A_C$ (the subspaces for $C$ and $D$ are orthogonal), the bias altering term and the dampening term are both 0. This means that WRING does not amplify the bias *at all* for orthogonal subspaces, unlike projection which always increases the bias for these spaces.

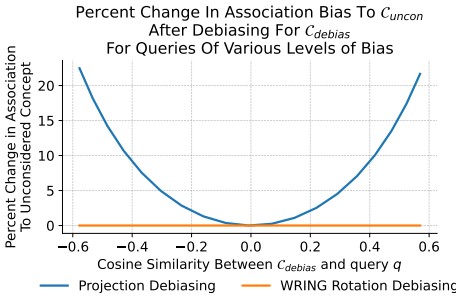
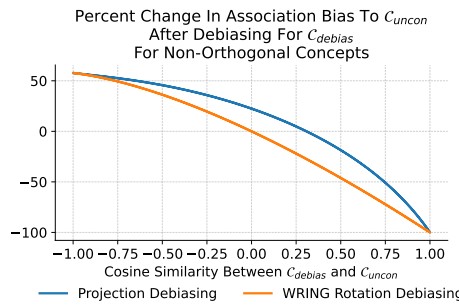
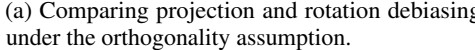

(a) Comparing projection and rotation debiasing under the orthogonality assumption.

(b) Comparing projection and rotation debiasing when orthogonality is not satisfied.

Figure 2: Synthetic data experiment.

## 4.1 COMPARING DEBIASING APPROACHES EMPIRICALLY

Now let us compare projection and WRING rotation debiasing in an analysis on synthetic data. We define a 5-dimensional space $\mathcal{X}$ consisting of two concept subspaces $C$ and $D$, each two dimensional, and one *signal* direction $s$. Each concept contains 2 groups; e.g. $C = \{c_1, c_2\}$ and $D = \{d_1, d_2\}$. Let $\boldsymbol{A}_C \perp \boldsymbol{A}_D$. Define four vectors $\boldsymbol{x}_{i,j} = \boldsymbol{c}_i + \boldsymbol{d}_j + \boldsymbol{s}$ for $i \in \{1, 2\}$ and $j \in \{1, 2\}$. Let $\boldsymbol{q}$ be a query vector that is likewise a mix of the concept directions and the signal direction: $\boldsymbol{q} = \omega\boldsymbol{c}_1 + \omega\boldsymbol{b}_1 + \boldsymbol{s}$. If $\omega = 0$, then the query is perfectly aligned with the signal (modeling the case where the query vector is not biased towards any concept).

First, we vary $\omega$ from $-1$ to $1$, and measure the change in bias for $\boldsymbol{A}_D$ after debiasing $\boldsymbol{q}$ with respect to $\boldsymbol{A}_C$ with projection and rotation debiasing. Figure 2a validates that projection debiasing amplifies bias in the unconsidered concept while WRING does not. Further, this amplification increases as the association between the query and the concept directions increases.

Next, we study what happens when $\boldsymbol{A}_C \not\perp \boldsymbol{A}_D$, varying the angle between the vectors in $\boldsymbol{A}_C$ and $\boldsymbol{A}_D$. Figure 2b shows that projection consistently has a higher increase in bias than rotation. When bias is amplified, projection increases faster than rotation. When bias in the unconsidered concept is inadvertently minimized, rotation minimizes the bias faster.

## 5 EXPERIMENTS

We evaluate the effectiveness of WRINGVLM at debiasing VLM embeddings while avoiding amplification of remaining biases. We compare WRINGVLM against prior debiasing methods across several benchmarks and experimental setups.

Table 1: Overview of the evaluation datasets. The last column shows the maximum average cosine similarity between directions in each concept space. Standard deviations given in the parenthesis.

| Dataset | Concepts | Query Types | Max. Concept Similarity |
|---|---|---|---|
| FairFace (Kärkkäinen and Joo, 2019) | {Gender, Race} | [Appearance, Behavior, Media Portrayal] | 0.0256 (0.0130) |
| CelebA (Liu et al., 2015) | {Gender, Race} | [Appearance, Behavior, Media Portrayal] | 0.0256 (0.0130) |
| Spawrious (Lynch et al., 2023) | {Dog Breed, Background} | [Dangerous, Protective, Friendly] | 0.0239 (0.0392) |
| Fashion (Aggarwal, 2019) | {Season, Color, Gender} | [Appearance] | 0.2309 (0.2216) |

## 5.1 EXPERIMENTAL SETUP

**Datasets.** We compare our WRINGVLM approach to existing debiasing approaches on the FAIRFACE (Kärkkäinen and Joo, 2019), CELEBA (Liu et al., 2015), SPAWRIOUS (Lynch et al., 2023), and FASHIONPRODUCTIMAGES (Aggarwal, 2019) datasets.

FAIRFACE and CELEBA consist of images of individual people, and we evaluate these datasets for race and gender bias. SPAWRIOUS consists of images of dogs of various breeds in different

locations. We evaluate SPAWRIOUS for `breed` and `background` bias. FASHIONPRODUCTIMAGES is evaluated for `season`, `color` and `gender` bias. Table 1 provides a dataset summary.

**Models.** We test on three widely used VLMs: `CLIP-ViT-B/32` (BP32) (Radford et al., 2021), `CLIP-ViT-L/14` (LP14) (Radford et al., 2021) and `CLIP-ViT-L/14-laion2B-s32B-b82K` (L162b) (Schuhmann et al., 2022). All models are evaluated in their frozen, pretrained forms without fine-tuning.

Figure 3: Comparison of debiasing approaches across models and datasets.

**Compared Methods.** We compare WRINGVLM against the following debiasing methods:

- **Baseline CLIP** (Radford et al., 2021). The original CLIP model (e.g. ViT-B-P16 or ViT-L-P14) without any debiasing steps.
- **Orthogonal Projection using Text Directions (Proj (txt))** (Chuang et al., 2023a). The `concept` directions are identified via text embeddings, and are removed via projection.
- **Orthogonal Projection using Image Directions (Proj (img))**. The `concept` directions are identified via image embeddings, and are removed via projection. We define the `group` direction as the $k = 100$ images with the highest similarity to the query, where these images are taken from a held-out reference dataset.
- **Selective Feature Imputation for Debiasing (SFID)** (Jung et al., 2024). `Concept` directions are identified by selecting the dimensions with the highest feature importance according to a Random Forest classifier trained to distinguish between groups. The values for these most important features are replaced with the average value of the features of low-confidence samples.

Additional implementation details are available in the Appendix.

## 5.2 EVALUATING BIAS AMPLIFICATION

We study the effect that the choice of debiasing operation has on amplifying bias by evaluating projection, selective feature imputation (SFID), and rotation (WRINGVLM) on the set of evaluation datasets. For each experiment, we select one concept to be debiased for ($\mathcal{C}_{debias}$) and one concept to be the *unconsidered* concept that was not debiased ($\mathcal{C}_{uncon}$). Ideally, the change in bias for the concept that was not debiased for should be zero. We thus examine the percent change in bias for the unconsidered concept $\mathcal{C}_{uncon}$, after debiasing for $\mathcal{C}_{debias}$. However, it is of course also important that the debiasing operation succeeds in eliminating bias for the target concept $\mathcal{C}_{debias}$. We thus report demographic parity for the worst group in $\mathcal{C}_{debias}$ as well.

We evaluate the human image datasets on word lists relating to appearance, behavior, and media portrayal. Specifically, we use the word lists for these concepts from the SO-B-IT taxonomy (Hamidieh et al., 2024). For the images of dogs, we used word lists relating to dangerous, protective, and friendly behaviors associated with dogs. All word lists are given in full in the Appendix.

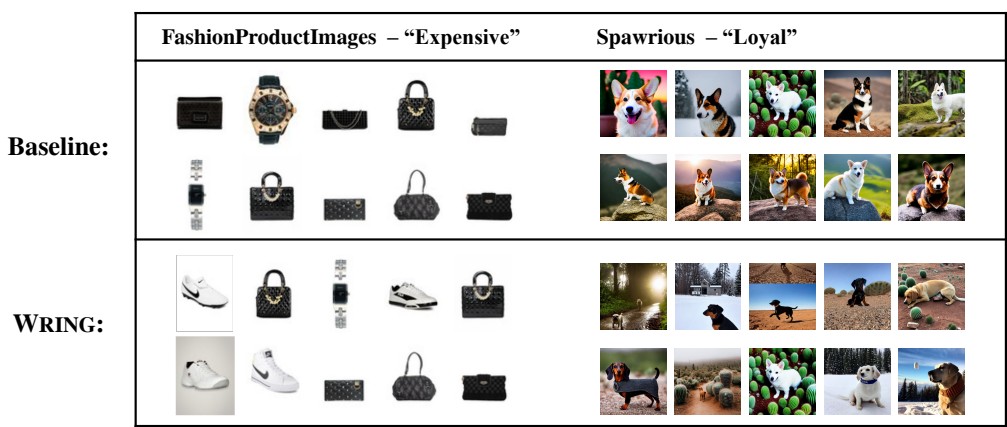

Figure 4: Top ten images for "expensive" items in `FashionProductImages` and "loyal" dogs in `Spawrious` before and after WRING debiasing. The sample after debiasing shows less color bias for the fashion images, and less bias towards corgis for the dog images. Fashion images were constrained to black/white images for demonstration. `clip-vit-base-patch32` backbone.

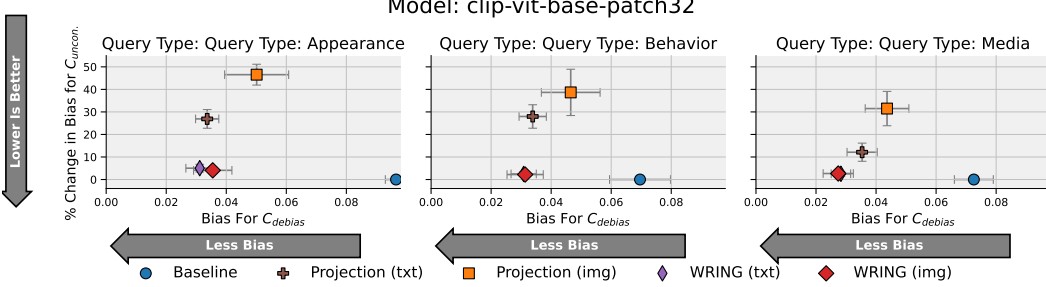

Figure 5: Projection vs rotation (WRING) for nonlinear debiasing on `FairFace`.

Figure 3 (a) shows a highly aggregated set of results, where for each dataset we average performance over backbone models, query type, and concept. We first note finding concept directions through embeddings of text descriptions (Projection (txt) and WRING (txt)) does not debias as well as when image embeddings are used to define these directions (Projection (img) and WRING (img)). While SFID typically does not show as much change in bias as projection, it also has the least decrease in bias for the target concept $\mathcal{C}_{debias}$. Thus, SFID does not show success in debiasing a target concept, even if it achieves little change in unconsidered directions. WRING exhibits much less bias amplification (% change in bias for $\mathcal{C}_{unconsidered}$) than projection, matching our theoretical claims.

Table 2: Worst Group Accuracy (higher is better) and Accuracy Gap (lower is better).

| Method | Worst Group Accuracy ↑ | | | | Accuracy Gap ↓ | | | |
|---|---|---|---|---|---|---|---|---|
| | Gender | Age | Skin Tone | Nose Shape | Gender | Age | Skin Tone | Nose Shape |
| Baseline | 72.78 | 87.42 | 86.36 | 85.88 | 17.02 | 1.57 | 5.30 | 5.04 |
| FairerCLIP | 84.78 | 82.66 | 83.33 | 84.21 | 11.71 | 13.49 | 12.75 | 13.47 |
| Projection | 78.89 | 85.14 | 85.30 | 84.55 | 11.87 | 5.05 | 6.36 | 7.33 |
| WRING | 80.56 | 85.59 | 85.00 | 85.22 | 9.24 | 4.07 | 8.88 | 5.22 |

We note that WRING has less *variance*, supporting our theoretical analysis: **projection doesn't *always* increase bias, but it's *unpredictable***. Figure 3 (b) shows results for each backbone model, averaging over datasets, concepts, and query types. We see that the findings from Figure 3 (a) still hold. Notably, we see that projection significantly changing bias when compared to WRING is consistent across the backbones. Figure 3 (c) shows least aggregated results, where we focus in on a single dataset (FairFace) and model (LP14). Again, the previous trends hold, though we see smaller variances than in the more aggregated results (additional results in the Appendix).

Overall, despite having minimal amplification of bias for unconsidered concepts, WRING still succeeds in debiasing the target concept. This can be seen by the relative performance along the horizontal axes in Figure 3, as well as visually in Figure 4.

## 5.3 REPLACING PROJECTION WITHIN OTHER DEBIASING PIPELINES

The projection operation is common in many debiasing approaches Wang et al. (2021); Chuang et al. (2023a); Seth et al. (2023); Zhang et al. (2024). As the previous experiment has shown, projection can introduce undetected bias amplification into pipelines that make use of that operation. We thus argue that projection can be swapped with WRING rotation to produce more stable debiasing pipelines.

To show the utility of replacing projection with WRING rotation, we focus on a specific debiasing pipeline that typically makes use of projection debiasing: *nonlinear* VLM debiasing Gerych et al. (2024). In this approach, the group directions are not assumed to be universal and are instead conditioned on the query. We evaluate projection and WRING rotation in this pipeline, where group directions are defined using text (txt) embeddings of the form `"a {group} {query}"` as well as using image directions (img). Figure 5 shows the relative performance of nonlinear VLM debiasing across backbones and query types, for the FairFace dataset. We show resuls for only the CLIP-VIT-BASE-PATCH32 model here, but include the full set of models in the appendix. We see that WRING rotation once again exhibits considerably less change in bias for the unconsidered concept than projection, while still succeeding to reduce bias for the target concept.

## 5.4 IMPACT ON DOWNSTREAM ACCURACY

We evaluate worst-group accuracy on a CELEBA hair-color task prediction, including an additional debiasing technique from the literature, FairerCLIP (Dehdashtian et al., 2024). We debias gender, and then evaluate worst group accuracy on a range of attributes. Results in Table 2 show that WRING improves target-concept worst-group accuracy, though FairerCLIP achieves the highest gains — but FairerCLIP requires labels and training, and is thus not directly applicable to our problem setting. However, we see that worst group performance for the unconsidered attributes is very close to the performance of the baseline for these groups, as expected, while FairerCLIP has consistently worse performance for the unconsidered concepts. This is in line with our main argument: WRING debiases the target concept without introducing hidden bias amplification for unconsidered concepts.

## 6 CONCLUSION

This work proposes a VLM debiasing method to mitigate bias across multiple different `concepts`, such as `race` and `gender`, by rotating the VLM embedding. By doing so, we mitigate the whac-a-mole effect - where removing bias along one `concept` axis via projection debiasing adds bias to a different `concept`. Our rotation debiasing method significantly decreases bias along target

`concept` axis while not increasing bias along other axes. Future work should continue to explore debiasing approaches with multiple concept considerations as VLM usage becomes increasingly common in real-world environments.

## 7 REPRODUCIBILITY STATEMENT

A description of our theoretical analysis, including derivations, is provided in Appendix A. Implementation details, such as compute resources, experimental setup, resampling procedures, and how group directions were defined using both text and image prompts, are provided in Appendix B. Full prompt lists and word queries are included in Appendix C. We evaluate across multiple publicly available datasets (FAIRFACE, CELEBA, SPAWRIOUS, and FASHIONPRODUCTIMAGES), each of which is fully described in Section 5.1. Additionally, we have uploaded our source code in the supplementary materials, which allows others to replicate our experiments and results.

## 8 ACKNOWLEDGMENTS

This work was supported in part by a National Science Foundation CAREER Award (2339381), AI2050 Award Early Career Fellowship (G-25-68042), Sloan Research Fellow Award (FG-2025-24277), Gordon & Betty Moore Foundation Award, and MIT-Google Computing Innovation Award. This material is based upon work supported by the National Science Foundation Graduate Research Fellowship under Grant No. (2141064).

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

## A   DERIVING THEORETICAL RESULTS

### A.1   DERIVING THE CHANGE IN BIAS AFTER PROJECTION

Equation 1 can be derived with simple linear algebra. First, consider the cosine similarity for the initial embedding $v$ and projection debiased embedding $v_{\backslash C}$:

$$\text{cosine\_sim}(v, d) = \frac{\langle v, d \rangle}{\|v\| \|d\|}$$

$$\text{cosine\_sim}(v_{\backslash C}, d) = \frac{\langle v, d \rangle - \langle P_C v, d \rangle}{\|v - P_C v\| \|d\|}$$

We can easily see how the bias changes for $d_1, d_2$:

$$\begin{aligned}
\text{bias}(v, d_1, d_2) &= \frac{\langle v, d_1 \rangle}{\|v\| \|d_1\|} - \frac{\langle v, d_2 \rangle}{\|v\| \|d_2\|} \\
&= \frac{\|d_2\| \langle v, d_1 \rangle - \|d_1\| \langle v, d_2 \rangle}{\|v\| \|d_1\| \|d_2\|} \\
&= \frac{\Delta_v}{\|v\| \|d_1\| \|d_2\|}, \Delta_v := \|d_2\| \langle v, d_1 \rangle - \|d_1\| \langle v, d_2 \rangle \quad \text{bias of original embedding}
\end{aligned}$$

$$\begin{aligned}
\text{bias}(v_{\backslash C}, d_1, d_2) &= \frac{\langle v, d_1 \rangle - \langle P_C v, d_1 \rangle}{\|v - P_C v\| \|d_1\|} - \frac{\langle v, d_2 \rangle - \langle P_C v, d_2 \rangle}{\|v - P_C v\| \|d_2\|} \\
&= \frac{\|d_2\| \langle v, d_1 \rangle - \|d_1\| \langle v, d_2 \rangle - \|d_2\| \langle P_C v, d_1 \rangle + \|d_1\| \langle P_C v, d_2 \rangle}{\|v - P_C v\| \|d_1\| \|d_2\|} \\
&= \frac{\|v\|}{\|v - P_C v\|} \cdot \frac{\Delta_v}{\|v\| \|d_1\| \|d_2\|} + \frac{\Delta_{P_C v}}{\|v - P_C v\|} \\
&= \underbrace{\frac{\|v\|}{\|v - P_C v\|}}_{\text{bias amplification}} \cdot \text{bias}(v, d_1, d_2) + \underbrace{\frac{\Delta_{P_C v}}{\|v - P_C v\|}}_{\text{bias altering}} \quad \text{bias after projection}
\end{aligned}$$

### A.2   OBTAINING $w$

We aim to construct a vector $w \in \text{col}(A_C)$ such that:

$$\langle w, c_i \rangle = \langle w, c_j \rangle \quad \text{for all } i, j = 1, \ldots, m,$$

where the unit vector $c_i$ is the $i$th column of $A_C$. This condition is equivalent to:

$$A_C^\top w \propto \mathbf{1},$$

where $\mathbf{1}$ is a vector of $m$ 1s.

Since $w \in \text{col}(A_C)$, we can express it as $w = A_C \alpha$ for some coefficient vector $\alpha \in \mathbb{R}^k$. Substituting this expression into the previous condition yields:

$$A_C^\top A_C \alpha \propto \mathbf{1}.$$

Since the columns of $A_C$ are linearly independent, $A_C^\top A_C$ is an $m \times m$ matrix with full rank. Thus, $A_C^\top A_C$ is invertible. We can thus find $\alpha$ as such:

$$\alpha = \left( A_C^\top A_C \right)^{-1} \mathbf{1}.$$

Substituting back, we get a vector $\tilde{w}$ that is is equivalent to the desired $w$ up to scale:

$$\tilde{w} = A_C \left( A_C^\top A_C \right)^{-1} \mathbf{1}.$$

The $w$ term in Equation 2 is thus:

$$w = \frac{\tilde{w}}{\|\tilde{w}\|}.$$

# B    IMPLEMENTATION DETAILS

## B.1    COMPUTE RESOURCES

Experiments were conducted using a remote compute cluster. We utilized an NVIDIA RTX A6000 GPU with 48 GB of VRAM, 60GB of RAM, and 8 processor cores. As the GPU was only required to obtain embeddings from pretrained CLIP models, a GPU with significantly less VRAM would have also sufficed. Results for (model, dataset) pairs typically took less than one hour.

## B.2    EXPERIMENTAL DETAILS

We run each experiment 100 times to obtain bootstrapped intervals. For each run, we resample the dataset with replacement. Next, we separate 20% of the bootstrapped dataset into a "reference" dataset, that is used to obtain the *image directions*. For each group, we find the mean embedding of the images of that group in the reference dataset to define the image directions. We use $K = 1000$. As CelebA did not include labels for race, we obtained pseuodlabels for white and black via CLIP zero-shot classification.

## B.3    DEFINING GROUP DIRECTIONS THROUGH TEXT PROMPTS.

To define group directions using texts prompts, we define a set of *group prompts* for each group, along with a set of *neutral prompts*. For instance, male prompts are of the form {a photo of a man, an image of a male, ...}, and neutral prompts are of the form {a photo of a person, an image of a person, ...}. We then embed each prompt in each set, and subtract the neutral embeddings from the group embeddings. We take the average of these differences to define the group text direction.

## B.4    METRICS.

We measure demographic parity:

$$\text{TopKAvg}(c) = \frac{1}{k} \sum_{s \in \text{Top}_k} \mathbb{I}[g(s) = c]$$

$$\text{DP}_{\text{top-}k}(C) = \max_{c \in C} \left| \text{TopKAvg}(c) - \frac{1}{|\mathcal{D}|} \sum_{s \in \mathcal{D}} \mathbb{I}[g(s) = c] \right|,$$

where $\mathcal{D}$ is a dataset and $g(s)$ is the group label for instance $s$.

We are also interested in how DP changes for unconsidered concepts. Define $\text{DP}'_{\text{top-}k}(C)$ as the DP after debiasing. Change in bias is then defined as:

$$\Delta\text{DP}(C) = \frac{|\text{DP}_{\text{top-}k}(C) - \text{DP}'_{\text{top-}k}(C)|}{\text{DP}_{\text{top-}k}(C)}$$

## C PROMPTS AND QUERIES

### C.1 PROMPTS FOR TEXT DIRECTIONS.

> **White Prompts**
> "A white person with blond hair is smiling.", "A white individual with freckles and red hair.", "The white person wears a baseball cap.", "A Caucasian individual sipping coffee by the window.", "A white child playing in the park.", "A white person with sunburned cheeks.", "A white individual with a thick beard and glasses.", "The white person adjusts their earrings.", "A white teen with braces and blue eyes.", "The white individual is wearing a flannel shirt.", "A white person jogging along the sidewalk.", "A Caucasian individual in a business suit.", "A white child with curly blonde hair.", "A white person with piercing green eyes.", "A white individual smiling with dimples.", "The white person leans against a brick wall.", "A white individual with straight brown hair.", "A white person in a hoodie and jeans.", "A white individual with a wide smile.", "A white person mid-laugh at a party.", "A white individual with fair skin and light eyes.", "A white person wearing a sunhat.", "A white person in a sports jersey.", "The white child carries a backpack.", "A Caucasian individual standing in the rain.", "The white person is applying lipstick.", "A white individual with slicked-back hair.", "A white person wearing a colorful scarf.", "A white individual adjusting their watch.", "A white person smiling for a photo.", "The white individual has a tattoo sleeve.", "A white person walking a dog.", "A white individual with light stubble.", "The white person wears a red coat.", "A white child with rosy cheeks.", "A white individual with a square jawline.", "A white person tilting their head slightly.", "A white individual raising their eyebrows.", "A Caucasian person standing by a window.", "A white person squinting in the sunlight.", "A white individual holding a coffee cup.", "The white child with their hair in a ponytail.", "A white individual with crowŽ2019s feet when they smile.", "A white person brushing their hair behind their ear.", "The white individual nods in conversation.", "A white child with wide blue eyes.", "A white individual wearing hoop earrings.", "A white person leaning on a railing.", "A white individual looking into the distance.", "The white person with blonde curls.", "A white individual buttoning their shirt.", "A white person with flushed cheeks.", "A Caucasian individual walking briskly.", "A white person sitting cross-legged.", "A white individual scratching their chin.", "A white person wearing glasses.", "A white individual with tousled hair.", "The white person smiling nervously.", "A white individual with arms crossed.", "A white person in a patterned dress."

**Black Prompts**
"A Black person with short, curly hair is smiling.", "A Black individual with dark skin and a broad nose.", "The Black person wears a baseball cap.", "An African-American individual sipping coffee by the window.", "A Black child playing in the park.", "A Black person with sun-kissed cheeks.", "A Black individual with a well-groomed beard and glasses.", "The Black person adjusts their earrings.", "A Black teen with a bright smile and dark eyes.", "The Black individual is wearing a denim jacket.", "A Black person jogging along the sidewalk.", "An African-American individual in a business suit.", "A Black child with coiled hair.", "A Black person with piercing brown eyes.", "A Black individual smiling with a gap-toothed grin.", "The Black person leans against a brick wall.", "A Black individual with long, straight hair.", "A Black person in casual clothes and sneakers.", "A Black individual with a friendly smile.", "A Black person mid-laugh at a party.", "A Black individual with dark skin and light eyes.", "A Black person wearing a sunhat.", "A Black person in a sports jersey.", "The Black child carries a backpack.", "An African-American individual standing in the rain.", "The Black person is applying lipstick.", "A Black individual with a stylish haircut.", "A Black person wearing a bright scarf.", "A Black individual adjusting their watch.", "A Black person smiling for a photo.", "The Black individual has a sleeve tattoo.", "A Black person walking a dog.", "A Black individual with light stubble.", "The Black person wears a leather jacket.", "A Black child with a bright smile.", "A Black individual with a chiseled jawline.", "A Black person tilting their head slightly.", "A Black individual raising their eyebrows.", "An African-American person standing by a window.", "A Black person squinting in the sunlight.", "A Black individual holding a coffee cup.", "The Black child with their hair in a ponytail.", "A Black individual with a joyful smile.", "A Black person brushing their hair behind their ear.", "The Black individual nods in conversation.", "A Black child with bright, wide eyes.", "A Black individual wearing hoop earrings.", "A Black person leaning on a railing.", "A Black individual looking into the distance.", "The Black person with curly hair.", "A Black individual buttoning their jacket.", "A Black person with a warm complexion.", "An African-American individual walking briskly.", "A Black person sitting cross-legged.", "A Black individual scratching their chin.", "A Black person wearing glasses.", "A Black individual with tousled hair.", "The Black person smiling softly.", "A Black individual with arms crossed.", "A Black person in a patterned shirt."

**Male Prompts**
"A man with a beard is smiling.", "He has short hair and a chiseled jawline.", "The young man is wearing a suit and tie.", "An older gentleman with gray hair.", "A muscular man flexing at the gym.", "A man with a five o'clock shadow.", "The boy wears a baseball cap.", "A clean-shaven man in a white shirt.", "The guy has a buzz cut.", "A father holding his child.", "A man in a leather jacket.", "A bearded man sipping coffee.", "The man has thick eyebrows.", "A confident man posing for the camera.", "The male model has a sharp look.", "A man with piercing blue eyes.", "The guy looks tough and athletic.", "A man in business attire.", "The man is smiling gently.", "A young man in a hoodie.", "A bald man with a goatee.", "The male soldier stands at attention.", "A teenage boy with messy hair.", "A male construction worker in a helmet.", "A man in a flannel shirt.", "A man with sideburns and stubble.", "The male athlete poses proudly.", "A male firefighter in uniform.", "A man with tired eyes.", "A male pilot in sunglasses.", "A man adjusting his collar.", "A boy with a playful grin.", "The man is mid-laugh.", "A man biting his lip.", "A man with a tattoo on his neck.", "A young man playing guitar.", "The man looks relaxed and confident.", "A guy in sportswear.", "The man is grinning.", "A man with a furrowed brow.", "The male soldier wears camouflage.", "A bearded man looking down.", "The guy has a pierced ear.", "A man in a trench coat.", "A man with broad shoulders.", "A man with a crooked smile.", "The dad has his baby on his shoulders.", "A man sipping from a mug.", "A man with round glasses.", "The male student has messy hair.", "The male hiker looks rugged.", "A guy with an intense gaze.", "His beard is thick and trimmed.", "A boy with freckles.", "A man walking in the city.", "The male mechanic is covered in grease.", "A man with a calm demeanor.", "A man brushing back his hair.", "A young man adjusting his glasses.", "A father cradling a newborn.", "A man with a stern gaze."

**`Female` Prompts**
"A woman with long hair is smiling.", "She has short hair and a soft jawline.", "The young woman is wearing a dress and heels.", "An older lady with gray hair.", "A fit woman stretching at the gym.", "A woman with a soft shadow of makeup.", "The girl wears a sun hat.", "A clean-faced woman in a white blouse.", "The lady has a sleek bob cut.", "A mother holding her child.", "A woman in a denim jacket.", "A woman with wavy hair sipping coffee.", "The woman has neatly shaped eyebrows.", "A confident woman posing for the camera.", "The female model has a striking look.", "A woman with piercing blue eyes.", "The lady looks graceful and athletic.", "A woman in professional attire.", "The woman is smiling gently.", "A young woman in a hoodie.", "A bald woman with large earrings.", "The female soldier stands at attention.", "A teenage girl with messy hair.", "A female construction worker in a hard hat.", "A woman in a flannel shirt.", "A woman with layered bangs and makeup.", "The female athlete poses proudly.", "A female firefighter in uniform.", "A woman with tired eyes.", "A female pilot in sunglasses.", "A woman adjusting her collar.", "A girl with a playful grin.", "The woman is mid-laugh.", "A woman biting her lip.", "A woman with a tattoo on her neck.", "A young woman playing guitar.", "The woman looks relaxed and confident.", "A girl in sportswear.", "The woman is grinning.", "A woman with a furrowed brow.", "The female soldier wears camouflage.", "A woman looking down with a braid.", "The girl has a pierced ear.", "A woman in a trench coat.", "A woman with broad shoulders.", "A woman with a crooked smile.", "The mom has her baby on her shoulders.", "A woman sipping from a mug.", "A woman with round glasses.", "The female student has messy hair.", "The female hiker looks adventurous.", "A girl with an intense gaze.", "Her curls are thick and bouncy.", "A girl with freckles.", "A woman walking in the city.", "The female mechanic is covered in grease.", "A woman with a calm demeanor.", "A woman brushing back her hair.", "A young woman adjusting her glasses.", "A mother cradling a newborn.", "A woman with a stern gaze."

**`Neutral` Prompts for `Gender and Race`**
"A person with a beard is smiling.", "Short hair and a defined jawline.", "A young adult wearing formal clothes.", "An older person with gray hair.", "An athletic person stretching at the gym.", "Someone with light facial hair.", "A person wearing a baseball cap.", "A person in a white shirt.", "A person with a short haircut.", "An adult holding a child.", "Someone in a leather jacket.", "A person sipping coffee.", "Thick eyebrows and a neutral expression.", "Someone posing confidently for the camera.", "A model with a striking look.", "Person with piercing blue eyes.", "An athletic person with toned features.", "An individual in business attire.", "A gentle smile on their face.", "A young person in a hoodie.", "A bald person with facial features.", "A soldier stands at attention.", "A teenager with messy hair.", "A construction worker in a helmet.", "Wearing a flannel shirt.", "Someone with styled hair and light makeup.", "An athlete posing proudly.", "A firefighter in uniform.", "Person with tired eyes.", "A pilot wearing sunglasses.", "Adjusting the collar of their shirt.", "A playful grin on their face.", "Laughing mid-sentence.", "Biting their lip slightly.", "A tattoo visible on the neck.", "Playing guitar outdoors.", "Relaxed and confident posture.", "Wearing sportswear.", "A bright grin across the face.", "A furrowed brow and intense gaze.", "In military fatigues, standing tall.", "Looking down with hair over the face.", "An earring glinting in one ear.", "Wearing a trench coat.", "Broad shoulders and neutral stance.", "A crooked smile forming.", "Carrying a child on their shoulders.", "Sipping from a mug.", "Wearing round glasses.", "A student with messy hair.", "A hiker with determined eyes.", "An intense, focused gaze.", "Thick curls around the face.", "Freckles across the cheeks.", "Walking in a busy city.", "A mechanic with grease-stained hands.", "A calm and quiet demeanor.", "Brushing hair out of the face.", "Adjusting their glasses.", "Cradling a newborn gently.", "A stern, focused look."

**`Dog Breed` Prompts [for Breed = {"bulldog", "corgi", "dachshund", "labrador"}]:**
"a photo of a {breed}", "an image of a {breed}", "a {breed}", "a picture of a {breed}"

> **Background Prompts [for Background = {"jungle", "mountain", "snow", "desert"}]:**
> 'a photo of a dog with a {background} background', "an image of a dog with a {background} background", "a dog with a {background} background", 'a picture of a dog with a {background} background'

> **Neutral Prompts for Background and Dog Breed:**
> 'a photo of a dog', "an image of a dog", "a dog", "a dog", 'a photo of my dog', "this dog", "my dog", "a puppy"

## C.2 QUERIES.

> **"Appearance" queries [FairFace, CelebA]:**
> attractive, beautiful, chubby, cute, disfigured, elegant, fat, fit, glamorous, groomed, handicap
>
> **"Behavior" queries [FairFace, CelebA]:**
> aggressive, ambitious, amused, angry, anxious, ashamed, attached, bored, bossy, brave, breadwinner, calm, caring, committed, compassionate, confident, conformist, content, curious, dainty, dedicated, dependent, dignified, disgusted, docile, domestic, eccentric, emotional, excited, friendly, frustrated, gentle, happy, helpful, hopeful, hopeless, hysterical, immature, independent, indifferent, insecure, intelligent, lazy, loud, loving, loyal, naggy, normal, nurturing, outcast, outgoing, passive, polite, popular, possessive, promiscuous, proud, rational, rowdy, sad, sensitive, shy, strong, studious, submissive, subservient, supportive, surprised, unapproachable, uncare, unconventional, unfriendly, unruly, untrustworthy, warm, weak, well-spoken
>
> **"Media Portrayal" queries [FairFace, CelebA]:**
> backward, barbaric, cosmopolitan, exotic, foreign, ghetto, hypersexual, immigrant, local, mysterious, native, outsider, primitive, rural, sassy, savage, third-world, traditional, tribal, uncivilized, uncultured, unrefined, urban
>
> **"Dangerous" queries [Spawrious]:**
> threatening a child, attacking someone, lunging forward
>
> **"Protective" queries [Spawrious]:**
> protecting a child, protecting someone, guarding
>
> **"Playful" queries [Spawrious]:**
> playing with a child, playing fetch, wagging tail

## C.3 UTILITY AFTER DEBIASING

Table 3: Accuracy after debiasing.

| Method | BP32 | | | LP14 | | | L14-LAION-2B | | |
|---|---|---|---|---|---|---|---|---|---|
| | Color | Gender | Season | Color | Gender | Season | Color | Gender | Season |
| **Topwear** | | | | | | | | | |
| Base | $0.438 \pm 0.010$ | $0.438 \pm 0.010$ | $0.438 \pm 0.010$ | $0.600 \pm 0.102$ | $0.597 \pm 0.102$ | $0.597 \pm 0.102$ | $0.813 \pm 0.087$ | $0.813 \pm 0.087$ | $0.813 \pm 0.087$ |
| Proj (txt) | $0.445 \pm 0.100$ | $0.491 \pm 0.092$ | $0.432 \pm 0.093$ | $0.558 \pm 0.109$ | $0.518 \pm 0.109$ | $0.581 \pm 0.092$ | $0.809 \pm 0.085$ | $0.803 \pm 0.102$ | $0.803 \pm 0.102$ |
| Proj (img) | $0.621 \pm 0.087$ | $0.859 \pm 0.072$ | $0.423 \pm 0.104$ | $0.539 \pm 0.096$ | $0.438 \pm 0.101$ | $0.608 \pm 0.096$ | $0.771 \pm 0.097$ | $0.793 \pm 0.089$ | $0.807 \pm 0.085$ |
| Wring (txt) | $0.322 \pm 0.091$ | $0.581 \pm 0.102$ | $0.141 \pm 0.066$ | $0.555 \pm 0.094$ | $0.321 \pm 0.081$ | $0.463 \pm 0.010$ | $0.803 \pm 0.092$ | $0.712 \pm 0.105$ | $0.834 \pm 0.082$ |
| Wring (img) | $0.448 \pm 0.101$ | $0.524 \pm 0.010$ | $0.435 \pm 0.094$ | $0.619 \pm 0.101$ | $0.562 \pm 0.102$ | $0.579 \pm 0.111$ | $0.632 \pm 0.109$ | $0.825 \pm 0.083$ | $0.808 \pm 0.096$ |
| **Bottomwear** | | | | | | | | | |
| Base | $0.257 \pm 0.080$ | $0.257 \pm 0.080$ | $0.257 \pm 0.080$ | $0.105 \pm 0.055$ | $0.105 \pm 0.055$ | $0.105 \pm 0.055$ | $0.636 \pm 0.103$ | $0.636 \pm 0.103$ | $0.636 \pm 0.103$ |
| Proj (txt) | $0.271 \pm 0.074$ | $0.250 \pm 0.077$ | $0.261 \pm 0.080$ | $0.102 \pm 0.054$ | $0.097 \pm 0.057$ | $0.110 \pm 0.058$ | $0.630 \pm 0.104$ | $0.602 \pm 0.107$ | $0.669 \pm 0.010$ |
| Proj (img) | $0.174 \pm 0.074$ | $0.113 \pm 0.064$ | $0.199 \pm 0.070$ | $0.102 \pm 0.058$ | $0.125 \pm 0.067$ | $0.097 \pm 0.059$ | $0.547 \pm 0.121$ | $0.564 \pm 0.114$ | $0.620 \pm 0.107$ |
| Wring (txt) | $0.326 \pm 0.090$ | $0.202 \pm 0.092$ | $0.460 \pm 0.083$ | $0.104 \pm 0.055$ | $0.102 \pm 0.060$ | $0.076 \pm 0.060$ | $0.596 \pm 0.103$ | $0.650 \pm 0.098$ | $0.670 \pm 0.098$ |
| Wring (img) | $0.274 \pm 0.079$ | $0.250 \pm 0.077$ | $0.279 \pm 0.083$ | $0.096 \pm 0.052$ | $0.094 \pm 0.054$ | $0.113 \pm 0.061$ | $0.628 \pm 0.108$ | $0.615 \pm 0.112$ | $0.692 \pm 0.091$ |

Table 3 shows the retrieval accuracy for "Topwear" and "Bottomwear" from the fashion image dataset before and after debiasing. WRING consistently maintains accuracy, showing that our proposed approach does not lead to a degradation of performance in downstream tasks.

## C.4 Disaggregated Results

Table 4: Bias in FAIRFACE for $\mathcal{C}_{debias}$ and $\mathcal{C}_{uncon}$ after debiasing for $\mathcal{C}_{debias}$. r = race, g = gender. Results shown for image directions.

| Model | Query | $\mathcal{C}_{debias}$ | $\mathcal{C}_{eval}$ | Change in Bias for Unconsidered Concept $\times 10^{-2}$ | | | Demographic Parity $\times 10^{-2}$ | | | |
| | | | | Proj | SFID | WRING | Baseline | Proj | SFID | WRING |
|---|---|---|---|---|---|---|---|---|---|---|
| LP14 | appearance | g | r | 2.36 ± 1.48* | 1.48 ± 2.71 | **0.17 ± 0.27** | 8.35 ± 0.60 | **2.10 ± 1.24*** | 6.33 ± 2.36 | 2.35 ± 1.66* |
| | appearance | r | g | 1.62 ± 0.98* | 0.60 ± 1.03 | **0.15 ± 0.22** | 6.67 ± 1.06 | **2.12 ± 1.04*** | 5.30 ± 2.06 | 2.52 ± 1.72* |
| | behavior | g | r | 0.84 ± 0.95 | 0.82 ± 1.22 | **0.06 ± 0.10** | 5.31 ± 0.59 | **1.99 ± 1.13*** | 4.83 ± 2.12 | 2.27 ± 1.60* |
| | behavior | r | g | 2.52 ± 1.45* | 0.94 ± 1.28 | **0.09 ± 0.13** | 6.00 ± 0.37 | **2.12 ± 0.87*** | 6.28 ± 2.05 | 2.59 ± 1.95* |
| | media | g | r | 0.98 ± 0.65* | 0.69 ± 0.88 | **0.63 ± 0.51*** | 9.23 ± 0.99 | **2.03 ± 1.01*** | 4.56 ± 2.25* | 2.24 ± 1.08* |
| | media | r | g | 3.63 ± 1.51* | 1.33 ± 1.95 | **0.34 ± 0.34** | 7.13 ± 0.58 | **2.04 ± 1.11*** | 6.45 ± 3.17 | 2.52 ± 1.78* |
| BP32 | appearance | g | r | 3.44 ± 1.83* | 1.53 ± 1.80 | **0.39 ± 0.43** | 9.26 ± 0.55 | **2.10 ± 1.74*** | 7.04 ± 2.89 | 2.33 ± 1.76* |
| | appearance | r | g | 0.57 ± 0.71 | **0.49 ± 0.68** | 0.54 ± 0.49* | 10.01 ± 1.67 | 2.60 ± 2.19* | 8.54 ± 3.86 | **1.97 ± 1.50*** |
| | behavior | g | r | 2.00 ± 1.97* | 0.68 ± 1.17 | **0.13 ± 0.16** | 5.01 ± 0.41 | **2.02 ± 1.58*** | 6.79 ± 3.60 | 2.18 ± 1.54* |
| | behavior | r | g | 1.30 ± 1.42 | 0.78 ± 1.22 | **0.17 ± 0.22** | 8.98 ± 0.95 | 2.31 ± 1.95* | 10.28 ± 2.97 | **2.03 ± 1.34*** |
| | media | g | r | 1.65 ± 1.76 | 0.59 ± 0.97 | **0.17 ± 0.28** | 6.09 ± 0.89 | 2.08 ± 1.91* | 6.15 ± 3.28 | **1.89 ± 1.33*** |
| | media | r | g | 2.31 ± 1.60* | 1.36 ± 1.48 | **0.22 ± 0.25** | 8.52 ± 0.98 | 2.45 ± 2.02* | 8.70 ± 2.75 | **2.06 ± 1.35*** |
| L162b | appearance | g | r | 1.64 ± 1.60* | 2.60 ± 1.53* | **0.47 ± 0.40*** | 9.71 ± 0.53 | **1.97 ± 1.32*** | 6.72 ± 4.13 | 2.14 ± 1.24* |
| | appearance | r | g | 1.52 ± 0.73* | 0.57 ± 0.65 | **0.56 ± 0.38*** | 11.91 ± 1.12 | 2.32 ± 1.34* | 12.17 ± 2.81 | **2.15 ± 1.04*** |
| | behavior | g | r | 1.33 ± 1.61 | 1.13 ± 0.88* | **0.11 ± 0.15** | 6.69 ± 0.53 | **2.02 ± 1.57*** | 5.83 ± 2.76 | 2.10 ± 1.26* |
| | behavior | r | g | 3.36 ± 1.67* | 0.68 ± 1.10 | **0.13 ± 0.19** | 10.17 ± 0.97 | 2.22 ± 1.41* | 9.59 ± 3.17 | **2.11 ± 1.01*** |
| | media | g | r | 0.63 ± 0.77 | 0.44 ± 0.65 | **0.24 ± 0.33** | 6.56 ± 0.53 | **1.98 ± 0.93*** | 6.34 ± 2.65 | 2.04 ± 1.29* |
| | media | r | g | 2.22 ± 1.59* | 0.43 ± 0.51 | **0.31 ± 0.36** | 10.69 ± 0.94 | 2.43 ± 1.48* | 11.18 ± 2.64 | **2.09 ± 0.95*** |

Table 5: Bias in CELEBA for $\mathcal{C}_{debias}$ and $\mathcal{C}_{uncon}$ after debiasing for $\mathcal{C}_{debias}$. r = race, g = gender. Results shown for image directions.

| Model | Query | $\mathcal{C}_{debias}$ | $\mathcal{C}_{eval}$ | Change in Bias for Unconsidered Concept $\times 10^{-2}$ | | | Demographic Parity $\times 10^{-2}$ | | | |
| | | | | Proj | SFID | WRING | Baseline | Proj | SFID | WRING |
|---|---|---|---|---|---|---|---|---|---|---|
| LP14 | appearance | g | r | 0.95 ± 0.65* | 1.32 ± 0.71* | **0.52 ± 0.49*** | 22.80 ± 0.67 | 7.62 ± 1.46* | 16.23 ± 2.90* | **6.91 ± 1.42*** |
| | appearance | r | g | 12.04 ± 1.54* | 3.45 ± 3.00* | **1.07 ± 0.76*** | 7.16 ± 0.60 | **4.46 ± 0.85*** | 7.33 ± 1.59 | 5.22 ± 1.47* |
| | behavior | g | r | 0.44 ± 0.48 | 0.71 ± 0.60* | 0.54 ± 0.39* | 17.70 ± 0.82 | 7.41 ± 1.70* | 12.23 ± 2.92* | **5.65 ± 0.87*** |
| | behavior | r | g | 20.52 ± 1.71* | 4.88 ± 5.20 | 0.53 ± 0.71 | 6.19 ± 0.40 | **4.32 ± 1.50*** | 6.99 ± 3.34 | 4.49 ± 1.52* |
| | media | g | r | **1.24 ± 0.79*** | 4.70 ± 0.98* | 2.37 ± 0.94* | 23.45 ± 0.86 | 8.85 ± 1.47* | 14.82 ± 2.88* | **8.47 ± 1.15*** |
| | media | r | g | 12.97 ± 2.11* | 3.50 ± 3.13* | 2.37 ± 0.94* | 11.71 ± 0.97 | 6.48 ± 1.67* | 9.53 ± 4.02 | **6.38 ± 1.85*** |
| BP32 | appearance | g | r | 1.71 ± 1.59* | 0.67 ± 0.53* | **0.58 ± 0.33*** | 22.82 ± 0.52 | **7.68 ± 1.14*** | 27.04 ± 4.36 | 8.54 ± 0.81* |
| | appearance | r | g | 9.88 ± 1.67* | 1.92 ± 4.37 | **0.27 ± 0.42** | 6.07 ± 0.55 | 6.19 ± 1.77 | 6.20 ± 1.02 | **3.18 ± 0.66*** |
| | behavior | g | r | 4.10 ± 1.93* | 1.07 ± 0.82* | **0.29 ± 0.20*** | 16.56 ± 0.52 | 7.49 ± 1.87* | 29.19 ± 7.68* | **6.15 ± 0.60*** |
| | behavior | r | g | 20.98 ± 2.64* | 6.96 ± 8.55 | **0.34 ± 0.44** | 6.76 ± 0.42 | 7.86 ± 2.55 | 5.08 ± 1.31* | **2.75 ± 1.00*** |
| | media | g | r | 3.30 ± 1.70* | 1.25 ± 1.23* | **0.37 ± 0.41** | 21.82 ± 0.56 | **6.87 ± 0.87*** | 24.21 ± 3.67 | 7.99 ± 1.14* |
| | media | r | g | 13.20 ± 2.14* | 1.87 ± 3.26 | **0.90 ± 0.77*** | 6.40 ± 0.50 | 7.00 ± 2.17 | 4.64 ± 0.84* | **3.29 ± 1.29*** |
| L162b | appearance | g | r | 2.07 ± 0.43* | 0.83 ± 0.69* | **0.10 ± 0.15** | 24.02 ± 0.57 | **7.49 ± 0.93*** | 12.86 ± 4.20* | 8.66 ± 0.90* |
| | appearance | r | g | 1.60 ± 1.13* | **0.63 ± 1.21** | 0.80 ± 0.51* | 5.92 ± 0.45 | 2.37 ± 0.61* | 5.80 ± 1.23 | **3.13 ± 0.45*** |
| | behavior | g | r | 2.55 ± 0.42* | 0.46 ± 0.49 | **0.31 ± 0.13*** | 17.34 ± 0.78 | 5.63 ± 0.82* | 13.97 ± 3.75 | **5.56 ± 0.48*** |
| | behavior | r | g | 16.27 ± 1.50* | 1.46 ± 1.89 | 1.06 ± 0.74* | 6.71 ± 0.40 | 2.21 ± 0.43* | 6.32 ± 0.97 | **3.15 ± 0.52*** |
| | media | g | r | 1.73 ± 0.48* | 0.73 ± 0.85 | **0.50 ± 0.25*** | 20.18 ± 0.64 | **8.37 ± 0.87*** | 18.84 ± 4.47 | 9.42 ± 0.74* |
| | media | r | g | 8.70 ± 1.34* | 0.93 ± 1.28 | **0.56 ± 0.45*** | 7.05 ± 0.46 | 2.32 ± 0.52* | 5.99 ± 0.86* | **3.03 ± 0.63*** |

Table 6: Bias in SPAWRIOUS for $\mathcal{C}_{debias}$ and $\mathcal{C}_{uncon}$ after debiasing for $\mathcal{C}_{debias}$. bg = background, br = breed. Results shown for image directions.

| Model | Query | $\mathcal{C}_{debias}$ | $\mathcal{C}_{eval}$ | Change in Bias for Unconsidered Concept $\times 10^{-2}$ | | | Demographic Parity $\times 10^{-2}$ | | | |
| | | | | Proj | SFID | WRING | Baseline | Proj | SFID | WRING |
|---|---|---|---|---|---|---|---|---|---|---|
| LP14 | friendly | br | bg | 1.25 ± 1.84 | **0.65 ± 1.05** | 1.07 ± 1.35 | 32.21 ± 2.27 | **5.95 ± 1.55*** | 32.92 ± 7.68 | 6.93 ± 1.99* |
| | friendly | bg | br | 7.38 ± 3.90* | 2.15 ± 2.96 | 3.18 ± 2.80* | 20.02 ± 1.51 | **5.18 ± 2.14*** | 19.15 ± 2.06 | 6.02 ± 1.80* |
| | protective | br | bg | 8.29 ± 2.37* | **0.89 ± 1.09** | 1.36 ± 1.83 | 52.43 ± 1.59 | **4.49 ± 1.87*** | 46.71 ± 7.64 | 5.00 ± 2.04* |
| | protective | bg | br | 1.51 ± 1.85 | 2.02 ± 2.64 | **0.77 ± 1.05** | 10.20 ± 1.39 | 7.87 ± 2.25* | 10.22 ± 2.72 | **7.29 ± 2.15*** |
| | dangerous | br | bg | 2.66 ± 2.19* | **0.78 ± 1.18** | 1.10 ± 1.47 | 32.49 ± 1.61 | 5.59 ± 2.29* | 30.84 ± 5.81 | **4.99 ± 2.18*** |
| | dangerous | bg | br | 8.03 ± 2.66* | **2.44 ± 2.75** | 4.76 ± 2.34* | 21.61 ± 1.64 | **5.67 ± 1.85*** | 21.70 ± 4.04 | 5.24 ± 1.82* |
| BP32 | friendly | br | bg | 5.47 ± 1.76* | **0.95 ± 1.53** | 2.31 ± 2.19* | 20.32 ± 1.95 | **4.26 ± 1.71*** | 22.06 ± 4.95 | 6.05 ± 2.15* |
| | friendly | bg | br | 4.36 ± 2.63* | **1.50 ± 2.42** | 1.59 ± 1.69 | 25.30 ± 1.61 | **4.63 ± 1.91*** | 30.29 ± 7.49 | 7.40 ± 2.66* |
| | protective | br | bg | 2.37 ± 2.94 | 1.55 ± 1.87 | **0.47 ± 0.73** | 13.16 ± 1.44 | **4.46 ± 1.62*** | 12.68 ± 2.38 | 7.54 ± 2.21* |
| | protective | bg | br | 3.37 ± 1.94* | 1.92 ± 2.02 | **0.70 ± 0.78** | 17.87 ± 1.72 | **4.44 ± 2.32*** | 23.57 ± 6.93 | 4.76 ± 1.80* |
| | dangerous | br | bg | 7.74 ± 2.42* | **2.44 ± 2.43*** | 3.87 ± 2.54* | 14.30 ± 1.35 | **5.21 ± 2.29*** | 14.89 ± 2.41 | 6.25 ± 2.28* |
| | dangerous | bg | br | 2.59 ± 1.73* | 1.15 ± 1.51 | **0.96 ± 0.94*** | 23.28 ± 2.00 | **4.19 ± 1.98*** | 25.96 ± 6.87 | 5.37 ± 1.60* |
| L162b | friendly | br | bg | **0.56 ± 0.77** | 0.88 ± 1.18 | 0.83 ± 0.77* | 14.11 ± 1.57 | **5.62 ± 1.89*** | 12.97 ± 2.84 | 6.06 ± 1.97* |
| | friendly | bg | br | 0.73 ± 1.07 | 0.97 ± 1.31 | **0.72 ± 1.11** | 20.18 ± 1.04 | **5.25 ± 1.32*** | 20.15 ± 3.88 | 6.25 ± 1.59* |
| | protective | br | bg | 4.08 ± 1.48* | 1.24 ± 1.64 | **0.82 ± 1.17** | 29.05 ± 1.65 | **4.87 ± 1.64*** | 23.18 ± 3.77* | 5.71 ± 2.03* |
| | protective | bg | br | 3.42 ± 1.74* | 1.59 ± 2.48 | **0.83 ± 1.12** | 19.39 ± 1.39 | **3.95 ± 1.72*** | 21.55 ± 6.81 | 4.47 ± 1.77* |
| | dangerous | br | bg | 2.41 ± 1.82* | **0.95 ± 1.32** | 1.38 ± 1.58 | 24.53 ± 1.54 | **4.39 ± 2.05*** | 24.42 ± 4.36 | 4.85 ± 2.28* |
| | dangerous | bg | br | 1.17 ± 1.21 | 1.39 ± 2.38 | **1.10 ± 1.25** | 16.95 ± 1.32 | **5.92 ± 1.75*** | 17.63 ± 4.88 | 6.05 ± 1.91* |

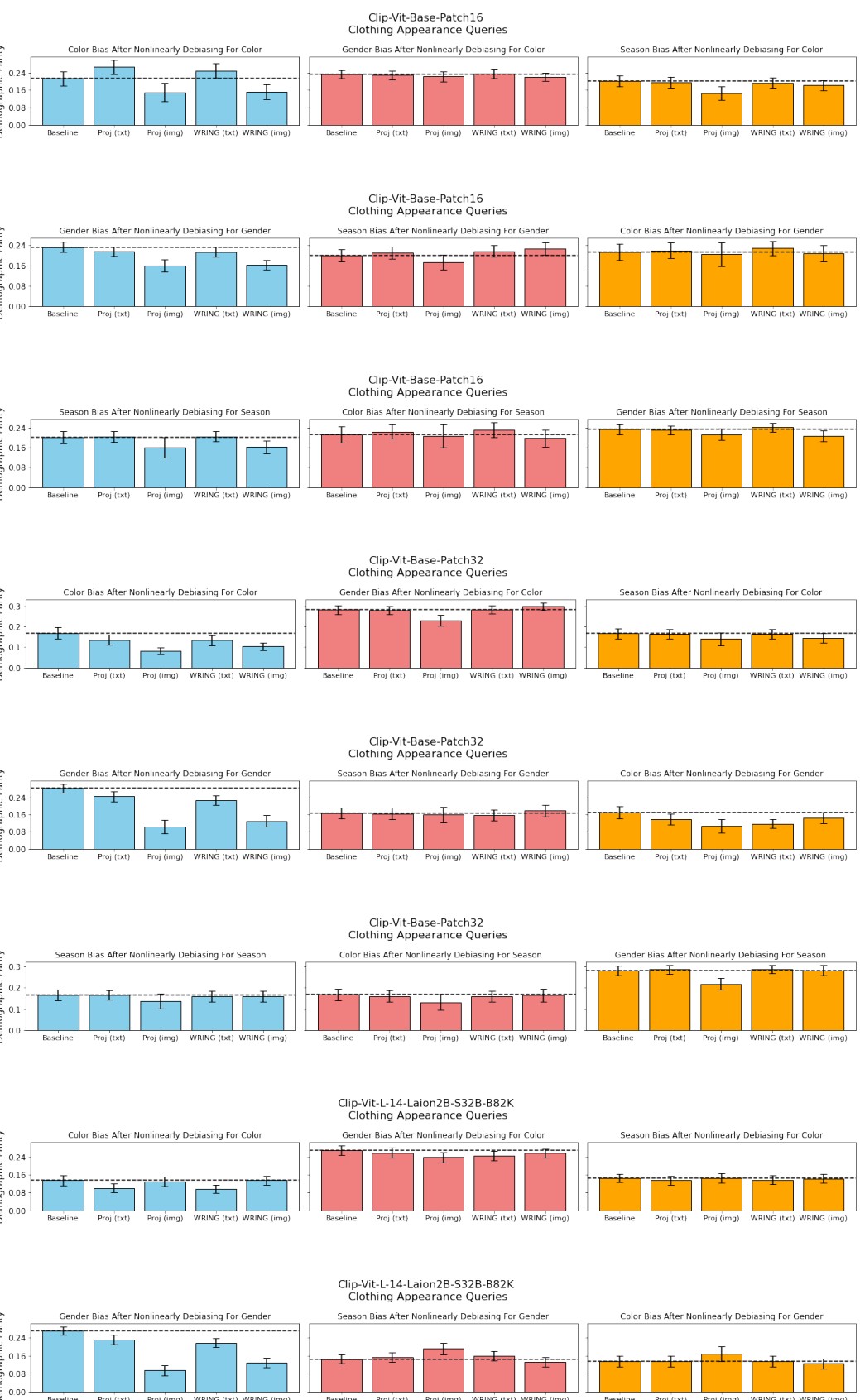

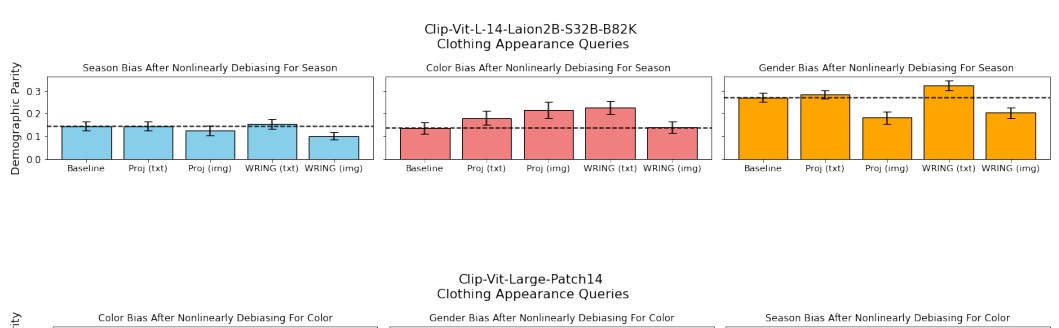

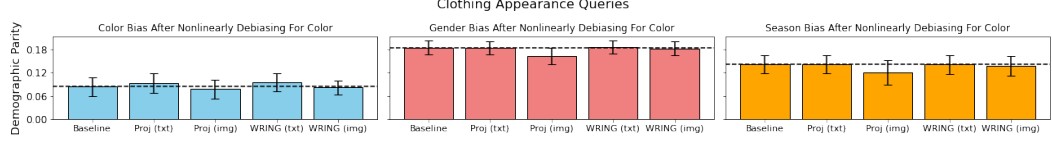

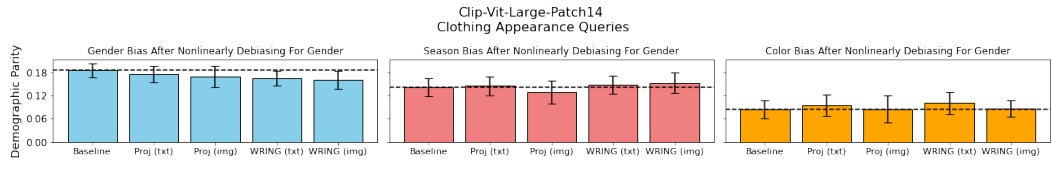

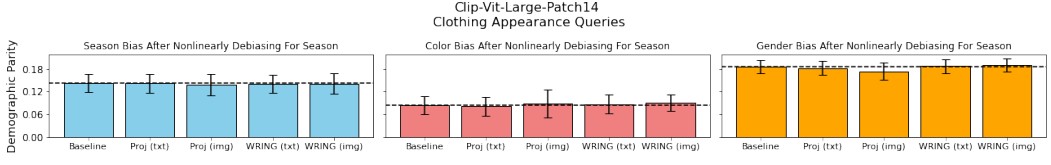

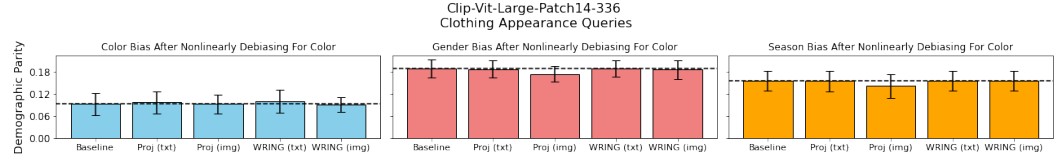

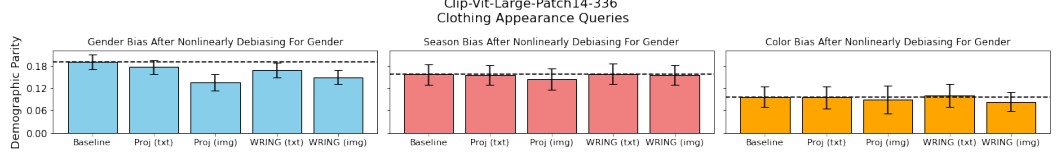

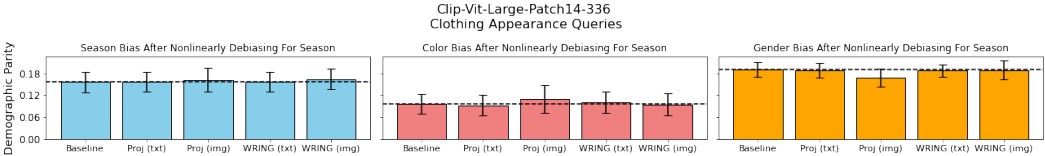

## C.5 Distributions

We perform an additional analysis on the distribution of cosine similarity between the `appearance` queries all `male` and `female` instances in `FairFace`. We see that both WRING and Projection almost always succeed in bringing the distribution of male and female images together after `gender` debiasing.

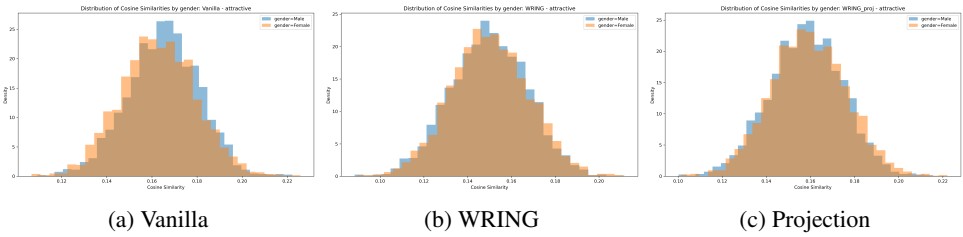

    (a) Vanilla               (b) WRING             (c) Projection

Figure 6: Gender split for "attractive"

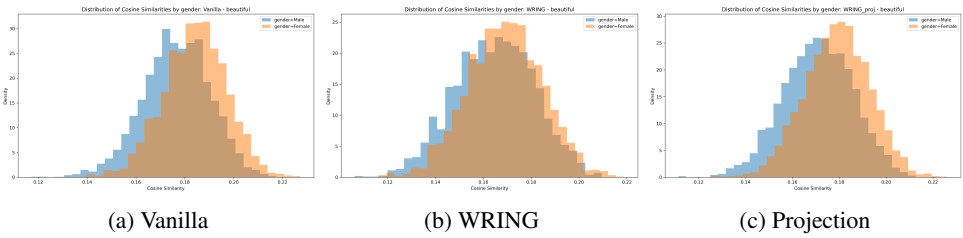

    (a) Vanilla               (b) WRING             (c) Projection

Figure 7: Gender split for "beautiful"

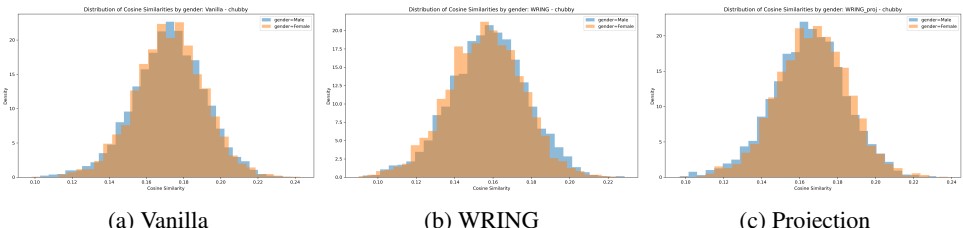

    (a) Vanilla               (b) WRING             (c) Projection

Figure 8: Gender split for "chubby"

## C.6 Signed Change in Bias

In addition to reporting the relative change in bias before and after debiasing, here we report the signed change in bias.

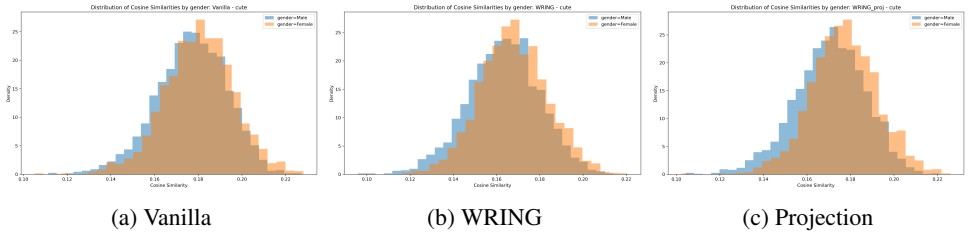

(a) Vanilla      (b) WRING      (c) Projection

Figure 9: Gender split for "cute"

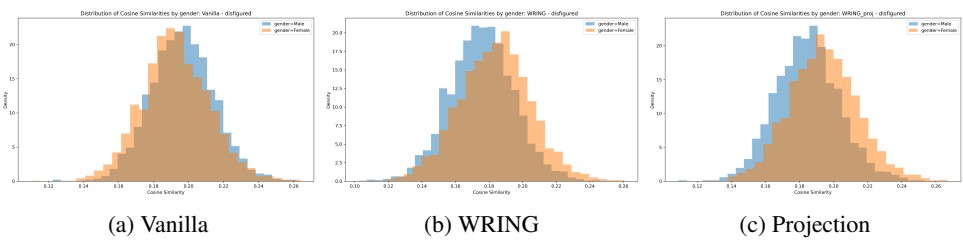

(a) Vanilla      (b) WRING      (c) Projection

Figure 10: Gender split for "disfigured"

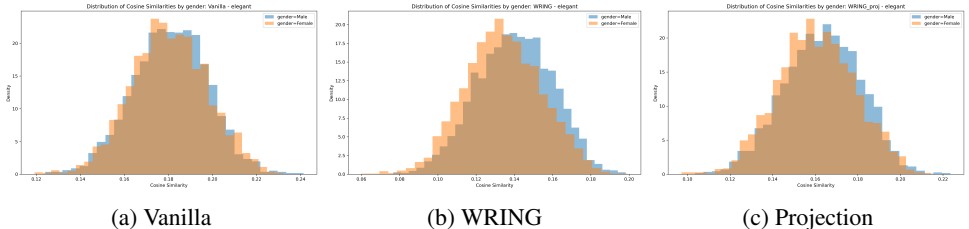

(a) Vanilla      (b) WRING      (c) Projection

Figure 11: Gender split for "elegant"

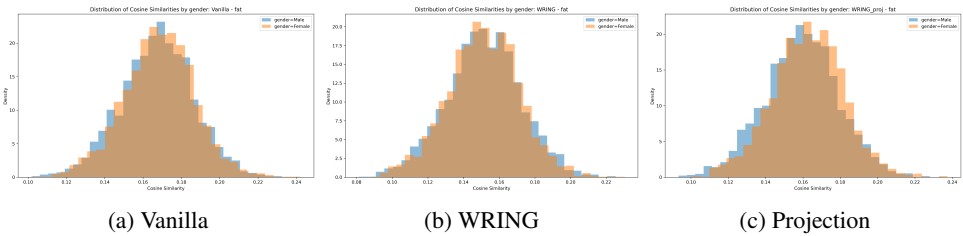

(a) Vanilla      (b) WRING      (c) Projection

Figure 12: Gender split for "fat"



(a) Vanilla      (b) WRING      (c) Projection

Figure 13: Gender split for "fit"



(a) Vanilla      (b) WRING      (c) Projection

Figure 14: Gender split for "glamorous"

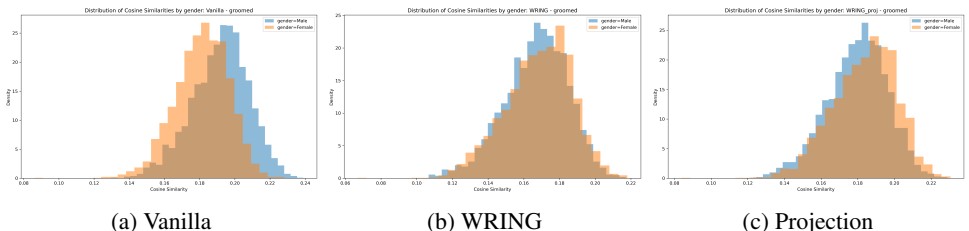

(a) Vanilla      (b) WRING      (c) Projection

Figure 15: Gender split for "groomed"

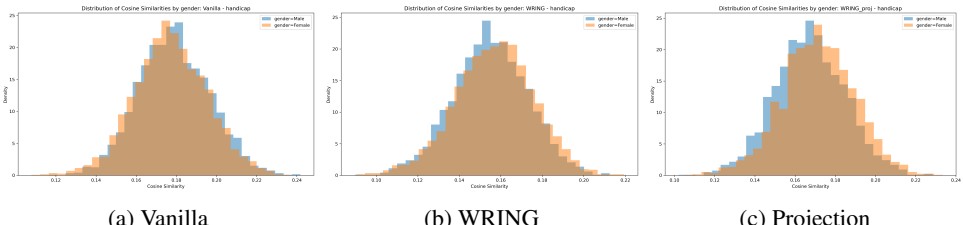

(a) Vanilla      (b) WRING      (c) Projection

Figure 16: Gender split for "handicap"

Table 7: Increase/Decrease in Bias. Model: `clip-vit-large-patch14`.

| Method | Bias for $C_{\mathbf{debias}} \downarrow$ | Change in bias for $C_{\mathbf{unconsidered}} \downarrow$ |
|---|---|---|
| Baseline | $0.167 \pm 0.0615$ | $0 \pm 0$ |
| Projection (txt) | $0.150 \pm 0.0563$ | $-0.594 \pm 2.51$ |
| Projection (img) | $0.0479 \pm 0.012$ | $26.7 \pm 17.4$ |
| SFID | $0.146 \pm 0.058$ | $4.77 \pm 9.04$ |
| WRING (txt) | $0.152 \pm 0.056$ | $-0.759 \pm 3.12$ |
| WRING (img) | $0.0484 \pm 0.0106$ | $0.441 \pm 4.31$ |

Table 8: Increase/Decrease in Bias. Model: `clip-vit-base-patch32`.

| Method | Bias for $C_{\mathbf{debias}} \downarrow$ | Change in bias for $C_{\mathbf{unconsidered}} \downarrow$ |
|---|---|---|
| Baseline | $0.135 \pm 0.0339$ | $0 \pm 0$ |
| Projection (txt) | $0.161 \pm 0.0609$ | $-0.307 \pm 3.91$ |
| Projection (img) | $0.0466 \pm 0.0113$ | $20.8 \pm 20.5$ |
| SFID | $0.152 \pm 0.0471$ | $1.39 \pm 9.06$ |
| WRING (txt) | $0.158 \pm 0.0546$ | $0.989 \pm 5.46$ |
| WRING (img) | $0.0454 \pm 0.0123$ | $-1.53 \pm 3.43$ |

Table 9: Increase/Decrease in Bias. Model: `CLIP-ViT-L-14-laion2B-s32B-b82K`.

| Method | Bias for $C_{\textbf{debias}}$ $\downarrow$ | Change in bias for $C_{\textbf{unconsidered}}$ $\downarrow$ |
|---|---|---|
| Baseline | $0.145 \pm 0.0351$ | $0 \pm 0$ |
| Projection (txt) | $0.162 \pm 0.0465$ | $0.596 \pm 1.71$ |
| Projection (img) | $0.0396 \pm 0.0106$ | $5.31 \pm 16.4$ |
| SFID | $0.131 \pm 0.0327$ | $-0.355 \pm 5.08$ |
| WRING (txt) | $0.153 \pm 0.0464$ | $0.304 \pm 2.58$ |
| WRING (img) | $0.0439 \pm 0.0118$ | $1.61 \pm 2.4$ |

