# OpenReview forum: "WRING Out The Bias: A Rotation-Based Alternative To Projection Debiasing"
_ICLR.cc/2026/Conference — ICLR 2026 Poster_

### Official Review · Reviewer_atPK · 2025-10-25

**Soundness:** 2
**Presentation:** 2
**Contribution:** 3
**Rating:** 6
**Confidence:** 4

**Summary:**

This paper introduces a modified debiasing mechanisms for vectorized representations, applied to vision language models (VLMs).  Its goal is to reduce the biased induced by a simple linear projection operation.  It does this by calculating some additional correlation that may happen with other identified concepts, and adjusting the vector representations to remove that additional correlation.

**Strengths:**

These are neat observations, and a clean mathematically justified approach on how best to adjust it.  The empirical results may (* see note below) show meaningful improvement, for a few simple cases in VLMs.

The approach and geometric modeling is nice and as far as I know new.  It is a subtle change to who debiasing uses projection that might (not sure) be the "right" way to do it to avoid unwanted bias in the process.

**Weaknesses:**

- the Bolukbasi etal 2016 paper, while introducing the idea to projection to remove bias, actually advocated for alternative approaches (e.g. Hard Debiasing and Soft Debiasing) which included other operations to try to correct for the affect of the projection on other identified concepts.  While the setting is different here (they considered word vectors), I think this paper should try to compare to the additional correction operations Bolukbasi proposed, and how they relate to the Wring operation.

  - Related to above, I believe the following paper is the one that actually first proposed just using the projection operation (without other corrections):  https://arxiv.org/abs/1901.07656

 - This is only applied to VLMs.  This is an important modern setting, but I wonder if it is applicable to text only or vision only models?  The reason to ask is that the text-focused debiasing does not seem to work as well in the experiments.

  - For most vectors v and concepts c, in high-dimensions the extra bias induced in projection should be small.  How targeted does this need to be to observe a meaningful difference?  For instance, in Sec 5.4 and Table 2 where you measure "Worst Group Accuracy" how many groups are there?  In general, I do not understand exactly what this is evaluating.

 - The main experiments are in Figure 3 & 5.  But I do not fully understand the x- and y-axis, which are labeled "Bias for C_debias" and "%Change in Bias for C_uncon."  These labels are not fully explained (or I missed it).  Yes a "bias" is defined in equations (1) and (3), but this takes in a vector v and two concepts d1, d2.  It is not clear (to me) how this equation is transformed into these the numbers reported on these axes.


Overall, I like the idea in this paper.  But because of the numerous questions above, especially in how to interpret the experimental results, I cannot yet stand fully behind it.

**Questions:**

See all questions in the weaknesses above.  The most critical ones are on understanding the precise things that were measured in the experiments.  I like aspects of this paper, but that needs to be clarified.

Second is understanding how this relates to methods actually discussed in Bolukbasi paper.  I do not think they had the same analysis as here, but the relation should be clarified.

**Details Of Ethics Concerns:**

This does clearly touch on bias issues.  But the set of techniques considered and modified are now rather standard.  I do not think there is much new in this paper that requires special attention given the existing methods.

---

> ### Author Response · Authors · 2025-11-26
>
> Thank you for your thoughtful review! We address your concerns below:
>
> # Comparison to extensions on projection debiasing
> We chose to limit the scope of this project to the theoretical application of projection debiasing which serves as a backbone to many different iterations of similar approaches. In all of these approaches, including the ones mentioned in Bolubasi et al., the central idea of debiasing remains the same: applying orthogonality to a considered concept direction. Our central argument is that this orthogonalization amplifies bias in unconsidered concepts, and that a rotational approach will not. We appreciate the additional citation mentioned and will include it in our camera-ready version.
>
> # Increase in bias in high dimensions
>
> Even if the change in magnitude between similarities of the query and group  embeddings is small in magnitude, what is important in the case of zeroshot retrieval and classification is the *relative* similarities. A small magnitude change in cosine similarity can still result in many more instances of one group being ranked higher than instances of another group. Our empirical results show that this increase in bias is significant (e.g. Figures 3 and 5) show that this increase in bias for the unconsidered concept is significant, even though the embedding models produce high dimensional embedding spacers.
>
> # Number of groups for each concept in Table 2
>
> Each concept in Table 2 has 2 groups.
>
> # Are these findings only applicable to VLM models?
>
> Our approach is applicable to embedding models that project inputs into a single shared vector space. WRING can indeed be applied to text-only or image-only models that produce a single vector embedding per input.
>
> # Unclear what metrics are benign reported in Figures 3&5
>
> We use max demographic parity as the bias metric: max | group_frequency_in_top_k - expected_group_frequency_in_top_k|, where "expected_group_frequency_in_top_k" is the group frequency of an ideal "fair" model. For *percent change* in bias, we compare the magnitude of the difference between bias (e.g. max demographic parity) before and after debiasing, normalized by the bias before debiasing: |bias_after - bias_before|/bias_before.

---

### Official Review · Reviewer_26nE · 2025-10-29

**Soundness:** 2
**Presentation:** 3
**Contribution:** 2
**Rating:** 4
**Confidence:** 4

**Summary:**

This paper introduces WRING, a method for debiasing the pre-trained representations of text queries from VLMs such as CLIP.  For a query embedding v and sensitive attribute embeddings $c_1$ and $c_2$, the authors define the bias of v with respect to $c_1$ and $c_2$ as:

bias(v, $c_1$, $c_2$) = cosine\_sim(v, $c_1$) - cosine\_sim(v, $c2$)

The method rotates the query embedding such that it becomes unbiased with respect to the considered set of attributes - bias(v, $c_1$, $c_2$) = 0. The authors theoretically show that WRING is less likely to amplify the bias (as per the author’s definition) of the query with respect to unconsidered sensitive attributes, than the classical projection-based debiasing approach.

**Strengths:**

1. The paper addresses a relevant topic - mitigating a set of known biases without amplifying unknown ones. The theoretical analysis seems correct.
2. The proposed procedure is not computationally expensive
3. The method is validated in diverse settings (multiple datasets and targets for debiasing), keeping in mind both the debiasing objective and the retention of performance for downstream tasks.

**Weaknesses:**

1. (minor) Limited scope \- the method is only applicable to CLIP-like models
2. The bias metric mentioned above seems to be particular to this work. While intuitive, the relevance of the metric itself is not supported through any arguments. The theoretical analysis on the change in bias is based on the author’s own metric for bias, and it is not argued that the same behaviour translates to established fairness metrics. Also, for the considered set of attributes there are multiple solutions for reducing this bias metric to 0 (e.g. Projection and WRING), yet they are unlikely to attain the same fairness performance, as measured through commonly accepted metrics.
3. The change in bias ($\\delta$DP) is the metric that highlights the differences between WRING in Projection debiasing \- I find them to perform reasonably similar with regards to the other metrics (Tables 2 and 3). However, $\\delta$DP does not reflect whether the change in bias was an increase or decrease. I think that not using the absolute value of the change in bias would better reflect this. If the change in bias brought by Projection is actually an improvement, that would oppose the claim of the first contribution and greatly reduce the value of the proposed method.

Initial recommendation: weak reject

Motivation: the article addresses a relevant problem, however, the first claim that Projection amplifies biases with regards to unconsidered attributes is insufficiently backed by the empirical results provided.

Additional feedback:

1. Reducing the bias usually comes at the cost of a drop in average performance. I would thus encourage the authors to also report the Average Accuracy in Table 2, or in the Appendix if there is not enough room for an additional column.
2. Tables 4-6 should also report the Baseline DP for unconsidered attributes to better put the absolute value for the change in bias into perspective.

Minor comments:

- The first 5-6 pages only refer to the bias metric introduced by the authors, but the metrics reported in all subsequent figures and tables are based on Demographic Parity (DP) instead. I would suggest adjusting Section 5.2 to make this aspect more clear.
- Line 247 \- in the definition of $\\delta\_w$, $d\_i$ and $d\_j$ appear instead of $d\_1$ and $d\_2$
- Line 250 \- bias(v, $d\_1$, \*\*$d\_1$\*\*)
- Appendix C.4 \- an explicit description of what the red color and the bold font represent would improve the readability of the tables.
- The figures on pages 22-23 do not seem to be properly introduced in the article. It is hard to tell at a glance what dataset was used and what is the setting

Things that could convince me to change my decision

- Addressing weakness 3 would be the most important and straightforward thing to do \- it just requires changing the evaluation metric and repeating the experiments for Projection. The change in bias for WRING is sufficiently low already.
- Having the Baseline DP for unconsidered attributes in Tables  4-6 would also help in better judging the impact of Projection-based debiasing on the unconsidered attributes.

Regarding weakness 2, using a surrogate metric for bias to motivate the approach is not necessarily an issue, and the analysis seems properly done. However, it would have been appreciated if the authors also motivated its use through some empirical evidence, e.g. higher bias translating into worse fairness, as judged by another metric.

**Questions:**

Can the authors address the following?

- Addressing weakness 3 would be the most important and straightforward thing to do \- it just requires changing the evaluation metric and repeating the experiments for Projection.
- Having the Baseline DP for unconsidered attributes in Tables  4-6 would also help in better judging the impact of Projection-based debiasing on the unconsidered attributes.

---

> ### Author Response · Authors · 2025-11-22
>
> Thank you for your constructive and insightful feedback. We address your concerns below.
>
> ### W1: (minor) the method is only applicable to CLIP-like models
>
> Our approach is indeed applicable to embedding models that project inputs into a single shared vector space, as is done in CLIP and CLIP-esque models. Extending our approach to decoder-only LLM-style embeddings, where each token is given its own embedding instead of a single vector representation for the entire input, is the next-step for our future work.
>
> ### W2: The bias metric mentioned above seems to be particular to this work.
>
> The metric used in our theoretical analysis is effectively just a measure of whether the query embedding is more similar to the embedding of one group over another. We are not trying to propose this as a new metric; rather, we focus on this aspect as any fairness measure that depends on similarity ordering (including standard retrieval metrics such as demographic parity and max skew) will inherit the same direction-dependent distortions. We report DP as it directly relates to the ordering-induced “metric” described in the earlier sections of the paper. We will update section 5 to make this more explicit.
>
> ### W3: Absolute Change in Bias vs Increase/Decrease in Bias
>
> Thank you for this thoughtful question. We report percent change in $C_{\text{unconsidered}}$ below, *without* taking any absolute values, in the tables below. We see that projection almost always increases bias for the unconsidered concepts, and when it does not it typically fails to significantly debias for the target concept. We will add these results to the updated paper.
>
>
> Model: clip-vit-large-patch14
>
> | Method | Bias for \(C_{\text{debias}}\) ↓ | Change in bias for \(C_{\text{unconsidered}}\) ↓ |
> |--------|-----------------------------------|----------------------------------------------------|
> | Baseline | 0.167 ± 0.0615 | 0 ± 0 |
> |---|---|---|
> | Projection (txt) | 0.15 ± 0.0563 | -0.594 ± 2.51 |
> | Projection (img) | 0.0479 ± 0.012 | 26.7 ± 17.4 |
> | SFID | 0.146 ± 0.058 | 4.77 ± 9.04 |
> | WRING (txt) | 0.152 ± 0.056 | -0.759 ± 3.12 |
> | WRING (img) | 0.0484 ± 0.0106 | 0.441 ± 4.31 |
>
> Model: clip-vit-base-patch32
>
> | Method | Bias for \(C_{\text{debias}}\) ↓ | Change in bias for \(C_{\text{unconsidered}}\) ↓ |
> |--------|-----------------------------------|----------------------------------------------------|
> | Baseline | 0.135 ± 0.0339 | 0 ± 0 |
> |---|---|---|
> | Projection (txt) | 0.161 ± 0.0609 | -0.307 ± 3.91 |
> | Projection (img) | 0.0466 ± 0.0113 | 20.8 ± 20.5 |
> | SFID | 0.152 ± 0.0471 | 1.39 ± 9.06 |
> | WRING (txt) | 0.158 ± 0.0546 | 0.989 ± 5.46 |
> | WRING (img) | 0.0454 ± 0.0123 | -1.53 ± 3.43 |
>
> Model: CLIP-ViT-L-14-laion2B-s32B-b82K
>
> | Method | Bias for \(C_{\text{debias}}\) ↓ | Change in bias for \(C_{\text{unconsidered}}\) ↓ |
> |--------|-----------------------------------|----------------------------------------------------|
> | Baseline | 0.145 ± 0.0351 | 0 ± 0 |
> |---|---|---|
> | Projection (txt) | 0.162 ± 0.0465 | 0.596 ± 1.71 |
> | Projection (img) | 0.0396 ± 0.0106 | 5.31 ± 16.4 |
> | SFID | 0.131 ± 0.0327 | -0.355 ± 5.08 |
> | WRING (txt) | 0.153 ± 0.0464 | 0.304 ± 2.58 |
> | WRING (img) | 0.0439 ± 0.0118 | 1.61 ± 2.4 |

---

> > ### Author Response · Authors · 2025-11-22
> >
> > ### Additional Feedback 1: Reducing the bias usually comes at the cost of a drop in average performance.
> >
> > Thank you for this suggestion. We have reported the accuracy on the Clothing dataset in Appendix C.3 in the current manuscript, but will extend this for all datasets.
> >
> > ### Additional Feedback 2: Adding baseline bias score for C_{unconsidered} for tables 4-6
> > We will add the baseline performance as suggested:
> >
> > Dataset:FairFace
> > | Model  | Query      | C_debias | C_eval | Baseline for C_{Uncon}     |
> > |--------|------------|----------|--------|------------------------|
> > | LP14   | appearance | g        | r      | 6.67 ± 1.06           |
> > | LP14   | appearance | r        | g      | 8.35 ± 0.60           |
> > | LP14   | behavior   | g        | r      | 6.00 ± 0.37           |
> > | LP14   | behavior   | r        | g      | 5.31 ± 0.59           |
> > | LP14   | media      | g        | r      | 7.13 ± 0.58           |
> > | LP14   | media      | r        | g      | 9.23 ± 0.99           |
> > |        |            |          |        |                        |
> > | BP32   | appearance | g        | r      | 10.01 ± 1.67          |
> > | BP32   | appearance | r        | g      | 9.26 ± 0.55           |
> > | BP32   | behavior   | g        | r      | 8.98 ± 0.95           |
> > | BP32   | behavior   | r        | g      | 5.01 ± 0.41           |
> > | BP32   | media      | g        | r      | 8.52 ± 0.98           |
> > | BP32   | media      | r        | g      | 6.09 ± 0.89           |
> > |        |            |          |        |                        |
> > | L162b  | appearance | g        | r      | 11.91 ± 1.12          |
> > | L162b  | appearance | r        | g      | 9.71 ± 0.53           |
> > | L162b  | behavior   | g        | r      | 10.17 ± 0.97          |
> > | L162b  | behavior   | r        | g      | 6.69 ± 0.63           |
> > | L162b  | media      | g        | r      | 10.69 ± 0.94          |
> > | L162b  | media      | r        | g      | 6.56 ± 0.53           |
> >
> > Dataset: CelebA
> > | Model  | Query      | C_debias | C_eval | FlippedBaseline      |
> > |--------|------------|----------|--------|------------------------|
> > | LP14   | appearance | g        | r      | 7.16 ± 0.60           |
> > | LP14   | appearance | r        | g      | 22.80 ± 0.67          |
> > | LP14   | behavior   | g        | r      | 6.19 ± 0.40           |
> > | LP14   | behavior   | r        | g      | 17.70 ± 0.82          |
> > | LP14   | media      | g        | r      | 11.71 ± 0.97          |
> > | LP14   | media      | r        | g      | 23.45 ± 0.86          |
> > |        |            |          |        |                        |
> > | BP32   | appearance | g        | r      | 6.07 ± 0.55           |
> > | BP32   | appearance | r        | g      | 22.82 ± 0.52          |
> > | BP32   | behavior   | g        | r      | 6.76 ± 0.42           |
> > | BP32   | behavior   | r        | g      | 16.56 ± 0.52          |
> > | BP32   | media      | g        | r      | 6.40 ± 0.50           |
> > | BP32   | media      | r        | g      | 21.82 ± 0.56          |
> > |        |            |          |        |                        |
> > | L162b  | appearance | g        | r      | 5.92 ± 0.45           |
> > | L162b  | appearance | r        | g      | 24.02 ± 0.57          |
> > | L162b  | behavior   | g        | r      | 6.71 ± 0.40           |
> > | L162b  | behavior   | r        | g      | 17.34 ± 0.78          |
> > | L162b  | media      | g        | r      | 7.05 ± 0.46           |
> > | L162b  | media      | r        | g      | 20.18 ± 0.64          |
> >
> > Dataset: Spawrious
> > | Model  | Query      | C_debias | C_eval | FlippedBaseline      |
> > |--------|------------|----------|--------|------------------------|
> > | LP14   | friendly   | br       | bg     | 20.02 ± 1.51          |
> > | LP14   | friendly   | bg       | br     | 32.21 ± 2.27          |
> > | LP14   | protective | br       | bg     | 10.20 ± 1.39          |
> > | LP14   | protective | bg       | br     | 52.43 ± 1.59          |
> > | LP14   | dangerous  | br       | bg     | 21.61 ± 1.64          |
> > | LP14   | dangerous  | bg       | br     | 32.49 ± 1.61          |
> > |        |            |          |        |                        |
> > | BP32   | friendly   | br       | bg     | 25.30 ± 1.61          |
> > | BP32   | friendly   | bg       | br     | 20.32 ± 1.95          |
> > | BP32   | protective | br       | bg     | 17.87 ± 1.72          |
> > | BP32   | protective | bg       | br     | 13.16 ± 1.44          |
> > | BP32   | dangerous  | br       | bg     | 23.28 ± 2.00          |
> > | BP32   | dangerous  | bg       | br     | 14.30 ± 1.35          |
> > |        |            |          |        |                        |
> > | L162b  | friendly   | br       | bg     | 20.18 ± 1.04          |
> > | L162b  | friendly   | bg       | br     | 14.11 ± 1.57          |
> > | L162b  | protective | br       | bg     | 19.39 ± 1.39          |
> > | L162b  | protective | bg       | br     | 29.05 ± 1.65          |
> > | L162b  | dangerous  | br       | bg     | 16.95 ± 1.32          |
> > | L162b  | dangerous  | bg       | br     | 24.53 ± 1.54          |

---

### Official Review · Reviewer_RKVo · 2025-11-03

**Soundness:** 3
**Presentation:** 3
**Contribution:** 3
**Rating:** 6
**Confidence:** 3

**Summary:**

This paper addresses the problem in VLMs that when debiasing for one concept via a projection, this can (often) amplify biases for other concepts. It starts by giving a geometric explanation of this phenomenon. Then it proposes an alternative debiasing technique, called WRING, that performs a rotation in the subspace of the to-be-debiased concept, instead of a projection. It gives a theoretical argument of why WRING suffers less from the above mentioned problem. Finally, it shows empirically that this is indeed the case: for various VLMs and datasets, it shows that WRING debiases for the concept-to-be-debiased, but has a less strong effect on the bias of other concepts that are not considered.

**Strengths:**

* WRING is, to the best of my knowledge, a novel approach in the field of post-hoc debiasing in VLMs. The novelty is in the fact that a rotation is being used. Since the problem the paper addresses is a relevant problem, a novel approach to tackling this problem is interesting and relevant for the scientific community.
* the paper gives a theoretical explanation of why the problem of bias amplification occurs, from a geometric point of view. This is insightful and also helps to understand the proposed method WRING.
* the experimental setting are mostly solid and the results are convincing.

**Weaknesses:**

* the theoretical analysis of the problem, in section 3, hints to a different solution than WRING. The bias amplification in Equation (1) is a consequence of the fact that an orthogonal projection reduces the length of the embedding vector. This can be addressed in different ways. One can simply rescale the projected embedding vector to its original length. Or one can use an oblique projection, where the range of the projection is chosen such that the length is invariant. The theoretical argument of why projection leads to bias amplification then disappears. It therefore would be very interesting and relevant to include a rescaling-after-orthogonal-projection method into the paper and see if WRING also outperforms such a method. If it does, it would also be very interesting to understand why. I very much would like to see such an analysis added to the paper. In the way the paper is written now, WRING does not seem the most natural or least impactful solution to the problem of bias amplification.
* there seems to be a mistake in the derivation of WRING. The condition in line 740 is not equivalent to the condition of unbiasedness for concept vectors $c_i$ and $c_j$ in line 227, if the length of those two vectors are unequal. I kindly ask the authors to clarify this and to check whether their WRING method is really doing what they intend it to do.
* all experiments are conducted on models that were not fine-tuned for their specific task. However, in practice, for many problems fine-tuning is a standard practice nowadays. It would therefore be highly relevant to also include experiments with fine-tuned models and to see if WRING then also outperform projection-based techniques.
* the analyses on pages 4 and 5 of the paper are assuming that the bias, defined in line 168, is always positive. But this is not true. Simply swapping the role of $c_1$ and $c_2$ flips the sign. I believe the text/analyses on pages 4 and 5 should be rewritten to make it consistent with this.
* the results in Table 2 would benefit from uncertainty estimates.

Some more minor weaknesses and typo's:
* there are quite a few language typo's. Please use some software to detect and correct them,
* line 247: indices $i$ and $j$?
* line 247: wrong symbol for length of vector (double lines).
* line 254: inequality sign wrong?
* line 257: "opposite". Is this always true? Why?
* line 269: $b$ -> $d$

**Questions:**

* in linear concept-removal techniques for classification models in the presence of spurious correlations, the concept-to-be-removed is often a binary label and is represented by a single direction in the embedding space. An example is the background in the Waterbirds dataset (land vs water). This seems to differ from VLMs, where the groups "land" and "water" would have their own vector in embedding space. What is the explanation of this difference? And suppose I would like treat the background as a single direction in the embedding space, how would WRING work in that case?
* how does the modality gap play a role in your analysis? It is not mentioned in the paper, although in the experiments there is a distinction between projections based on text-directions and image-directions.

---

> ### Author Response · Authors · 2025-11-26
>
> Thank you for your constructive and thoughtful review. We address your concerns below.
>
> ## Bias amplification as a result of decreasing the length of the vector embedding
> This is a reasonable misunderstanding, but it is not accurate. The bias amplification is not a result of the length of the vector changing. We measure bias in terms of cosine similarity, which is agnostic to the length of the vectors. Our results are equally valid in the case where vectors are renormalized. We will make this more clear in the camera ready version.
>
> ## Validity of derivation of WRING w.r.t. Line 740
> The condition stated in line 740 does match our definition of bias (the cosine similarity between query embedding and attribute embeddings being equal to each other) under the assumption that the query embedding and attribute embeddings have unit length. While we stated this assumption for the query embedding, we see that we did not state this assumption for the attribute embeddings. We will update our manuscript to state this explicitly. We note that this does not change the validity of our findings; we were working under the definition of attribute embeddings being normalized to have unit length, but failed to state this explicitly.
>
> ## Inclusion of fine-tuned models
> We believe that including a fine-tuned model is outside the scope of our work. The benefit of fine-tuning is largely to increase performance when you have knowledge of a predefined task. However, a central benefit of WRING is for tasks where there is minimal knowledge and you still want to ensure no increase in bias in unconsidered directions. We believe that this is the harder scenario for a debiasing method to work well in. However, we note that studying different debiasing methods and their effects on bias and accuracy under different types of models is an interesting area for future work.
>
> ## Direction of bias
> When defining bias on line 160, the sign of the bias indicates the direction. A higher similarity to ‘indoors’ in our example (as opposed to ‘outdoors’) would then give a positive bias in the direction of indoors. While you can swap the two vectors to flip the sign, the magnitude of the bias remains the same. Therefore, the remaining calculations on pages 4 and 5 remain consistent, just with a change in sign in the final result indicating directionality. We will update the text to clarify any inconsistencies.
>
> ## Modality Gap
> We find that image-based debiasing works better than just text-based debiasing. This means that having access to sets of images leads to a more effective debiasing approach than only having text. We hypothesize that this is due to a limited ability for text to capture concept directions.

---

### Meta-Review · Area_Chair_fXmX · 2026-01-07

**Summary:**

The paper proposes WRING, a rotation-based debiasing method for vision–language model embeddings that aims to reduce bias for known concepts without amplifying bias for unconsidered ones. Unlike projection-based debiasing, WRING preserves relationships to other concepts by rotating embeddings within a relevant subspace rather than removing components. Theoretical analysis and experiments across multiple VLMs and datasets show reduced unintended bias amplification while maintaining comparable debiasing effectiveness.

**Reviewer Concerns:**

None of the major concerns have not been addressed

**Reviewer Scores:**

6,4,6

---

### Decision · Program_Chairs · 2026-01-26

Accept (Poster)